# Variant-specific inflation factors for assessing population stratification at the phenotypic variance level

Tamar Sofer [1,2,3✉], Xiuwen Zheng [4], Cecelia A. Laurie [4], Stephanie M. Gogarten [4], Jennifer A. Brody [5], Matthew P. Conomos [4], Joshua C. Bis[4], Timothy A. Thornton[4], Adam Szpiro[4], Jeffrey R. O'Connell[6], Ethan M. Lange[7], Yan Gao[8], L. Adrienne Cupples [9,10], Bruce M. Psaty[5], NHLBI Trans-Omics for Precision Medicine (TOPMed) Consortium* & Kenneth M. Rice [4]

In modern Whole Genome Sequencing (WGS) epidemiological studies, participant-level data from multiple studies are often pooled and results are obtained from a single analysis. We consider the impact of differential phenotype variances by study, which we term 'variance stratification'. Unaccounted for, variance stratification can lead to both decreased statistical power, and increased false positives rates, depending on how allele frequencies, sample sizes, and phenotypic variances vary across the studies that are pooled. We develop a procedure to compute variant-specific inflation factors, and show how it can be used for diagnosis of genetic association analyses on pooled individual level data from multiple studies. We describe a WGS-appropriate analysis approach, implemented in freely-available software, which allows study-specific variances and thereby improves performance in practice. We illustrate the variance stratification problem, its solutions, and the proposed diagnostic procedure, in simulations and in data from the Trans-Omics for Precision Medicine Whole Genome Sequencing Program (TOPMed), used in association tests for hemoglobin concentrations and BMI.

[1] Division of Sleep and Circadian Disorders, Brigham and Women's Hospital, Boston, MA, USA. [2] Department of Medicine, Harvard Medical School, Boston, MA, USA. [3] Department of Biostatistics, Harvard T.H. Chan School of Public Health, Boston, MA, USA. [4] Department of Biostatistics, University of Washington, Seattle, WA, USA. [5] Cardiovascular Health Research Unit, Departments of Medicine, Epidemiology, and Health Services, University of Washington, Seattle, WA, USA. [6] Department of Medicine, Division of Endocrinology, Diabetes, and Nutrition, University of Maryland School of Medicine, Baltimore, MD, USA. [7] Department of Medicine, University of Colorado Anschutz Medical Campus, Aurora, CO, USA. [8] School of Medicine, University Mississippi Medical Center, Jackson, MS, USA. [9] Department of Biostatistics, Boston University School of Public Health, Boston, MA, USA. [10] National Heart, Lung, and Blood Institutes, Framingham Heart Study, Framingham, MA, USA. *A list of authors and their affiliations appears at the end of the paper. ✉email: tsofer@bwh.harvard.edu

Large-scale association analyses using whole genome sequence (WGS) data on thousands of participants are now underway, through programs such as the NHLBI's Trans-Omics for precision Medicine (TOPMed) program and NHGRI's Genome Sequencing Program. Unlike earlier Genome-Wide Association Studies (GWASs), where data were combined using meta-analyses of summary statistics, in WGS analyses participant-level data from multiple studies are often pooled, and results are obtained from a single analysis. Pooled analysis of WGS is useful due to its computational tractability and its ability to control for genetic relatedness across the pooled datasets. However, it is sensitive to a form of population stratification that is not well known. Population stratification in genetic association analysis[1,2] typically refers to situations where the mean phenotype value and allele frequency both differ across population subgroups. Unless appropriately accounted for in the analysis, e.g., by using regression-based adjustments for ancestry such as principal components or genetic relatedness matrices in linear mixed models, or their combination, it can lead to false–positive associations[3–5]. Population stratification more generally can refer to differences in phenotype distribution and allele frequency across population subgroups[6], and hence can also manifest as differences in phenotype variances by subgroup, combined with differences in allele frequencies. In practice, this phenomenon is common in pooled analysis of multi-study data, as small differences in allele frequencies are prevalent, and different studies being pooled often have different measurement protocols, environmental exposures and inclusion criteria, all of which can lead to different phenotype variances among studies.

Previous studies considered the effect of combining together groups with different phenotypic variances. Haldar and Ghosh[6] studied the effect of population stratification due to mean differences, variance differences, and more generally, phenotypic heterogeneity, across subpopulations, on false positive detections when testing variant associations with a quantitative trait. Conomos et al.[7] showed that when testing variant associations in a pooled sample of Hispanics/Latinos from different Hispanic background groups, statistical properties of tests are improved when the model allows for different residual variances in the different background groups. Musharoff et al.[8], in a preprint, studied population variance structure using statistical models of both population means and variances, and developed statistical tests for the association of genetic variants with phenotypic variability.

In this manuscript, we develop variant-specific inflation factors $\lambda_{vs}$, which quantify the degree of inflation/deflation in association testing of a single genetic variant due to population stratification at the variance level. We develop an algorithm to compute approximate variant-specific inflation factors based on allele frequencies and variances in groups pooled together for analysis, demonstrate their usage for assessing model fit, and demonstrate the implications of the population stratification at the variance level in simulations and in analyses of WGS data from TOPMed. To account for population stratification at the variance level we use the computationally efficient and scalable approach proposed in Conomos et al.[7] and implemented in GENESIS[9], and show in simulations that it indeed addresses the variance stratification problem in scenarios based on Musharoff et al.[8].

## Results

**Simulation studies.** Our simulations consisted of 576 simulation settings according to various combinations of parameters. We compared a few ways to estimate variance parameters to be used in computing $\lambda_{vs}$: empirical variances based on homogeneous and stratified variance models, and model-based variances from the

heterogeneous variance model. The estimated $\lambda_{vs}$ were essentially the same regardless of the method. Figure 1 compares the estimated $\lambda_{vs}$ to the observed $\lambda_{gc}$ in each of the simulation settings and in each of the two modeling approaches (homogeneous versus stratified variance). Settings are divided according to patterns determining whether variance stratification will be expected, including same or different MAFs between the two studies, the same or different error variances, and whether the PC affects the genetic variance or not. The top three rows in Fig. 1 demonstrate settings in which both the MAF and the total variances differ between the two combined studies, including settings in which both the error and genetic variance components are the same, but the PC affects the genetic variance, resulting in different total variance between the studies because the mean of the PC differs between them. In these settings the variance stratification is observed when using the homogeneous variance model, in that the observed inflation can be substantially higher or lower than 1, with exact values depending on the specific parameters used in each simulation. Indeed, the computed $\lambda_{vs}$ and the observed $\lambda_{gc}$ are highly correlated. In contrast, the stratified variance model was robust to variance stratification across all settings, with observed inflation around 1 in all simulations. The bottom two rows of Fig. 1 demonstrate settings in which either the MAF or the variances are the same in the two combined studies. In these settings, the expected inflation computed by $\lambda_{vs}$ is always 1 (no inflation). As expected, the observed inflation is the same in the homogeneous and stratified variance models. The spread seen in the values of the observed inflation, with some values higher and some lower than the desired 1, are consistent with that expected based on the number of replication of simulations in each setting (10,000); see Supplementary Information for more details.

**Genetic association analysis of BMI and hemoglobin concentration in TOPMed.** We demonstrate the variance stratification problem in analyzes of hemoglobin concentrations (HGB, $N = 7596$; three analysis groups) and body mass index (BMI, $N = 9807$; eight analysis groups) in the TOPMed freeze 4 dataset. In both analyses we computed approximate variant-specific inflation factors $\lambda_{vs}$. We investigated the inflation/deflation problems resulting from variance stratification, and verified that the patterns of inflation and deflation in the homogeneous variance analysis agree, across the different variants, with those obtained from the formula and the provided code. Figures 2 and 3 provide quantile-quantile (QQ)-plots for variants from three categories of variants, where theory predicts inflation ($\lambda_{vs} \geq 1.01$), deflation ($\lambda_{vs} \leq 0.99$) and variants with $\lambda_{vs}$ "Approx. no inflation" ($0.99 < \lambda_{vs} < 1.01$), and across all variants, for HGB and BMI analyses respectively. The plots overlay the results from the four analyses methods together. While the homogeneous variance model clearly produces inflated and deflated QQ-plots in line with the theoretical expectation, when looking at all tested variants together, this inflation and deflation (i.e., Type I and Type II errors) mask each other, alarmingly. Despite appearances, these problems do not "cancel out"; one creates more Type I errors, one creates more Type II errors, yet the plot of all results may lead investigators to conclude that the analysis is well-calibrated. In contrast, the stratified residual variance model provides good control of Type I errors, as seen in the QQ-plots, with the exception of the bottom left panel in Fig. 3, which provides QQ-plots for the set of variants that are expected to have deflated test statistics under the pooled variance model when studying BMI: here the stratified residual variance model was also somewhat deflated. Figure 4 provides the genomic control inflation factors $\lambda_{gc}$ computed over each of the variant sets provided in the QQ-plots and for each of the traits. The completely stratified and

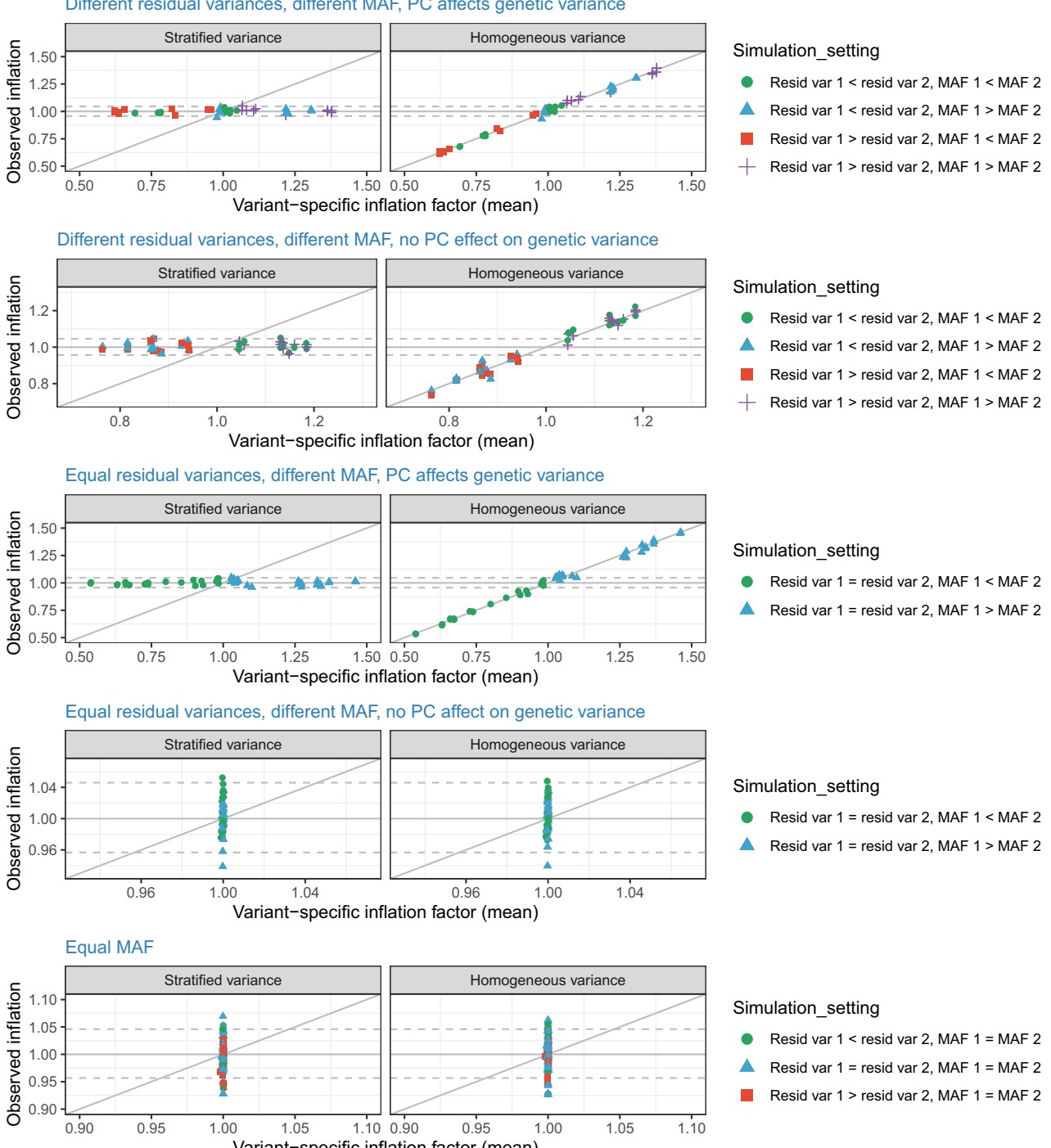

**Fig. 1 Estimated variant-specific inflation factors versus observed inflation in simulations.** The figure compares estimated variant-specific inflation factors $\lambda_{vs}$ estimated in each of many simulation settings, and corresponding observed inflation $\lambda_{gc}$ averaged across 10,000 repetitions of each simulation settings. Observed inflation values are provided based on a homogeneous variance model, in which a single variance parameter is estimated using the aggregated data; and based on a stratified variance model, that fits a different variance parameter to each of the two simulated studies. Each simulation set corresponds to a single point on this figure, and the simulations are grouped (denoted by different colors and symbols) by the characteristics stated in the legend. Within each group of simulation settings, the simulation parameters differ by specific parameter values, including MAFs, variance components, and sample sizes, while still satisfying the broad conditions of the grouped simulation settings. The dashed horizontal lines correspond to the 2.5% and 97% quantiles of the distribution of $\lambda_{gc}$ based on 10,000 variants under the null of no inflation/deflation, obtained from simulations.

MetaCor models performed better in terms of overall QQ-plots and computed $\lambda_{gc}$ values in the two analyses, in that $\lambda_{gc}$ values were always closer to 1. MetaCor performed slightly better than the completely stratified model under independence, likely because it accounts for a low degree of relatedness between the strata.

Table 1 describes the inflation/deflation patterns of variants according to their MAF. One can see that the inflation/deflation

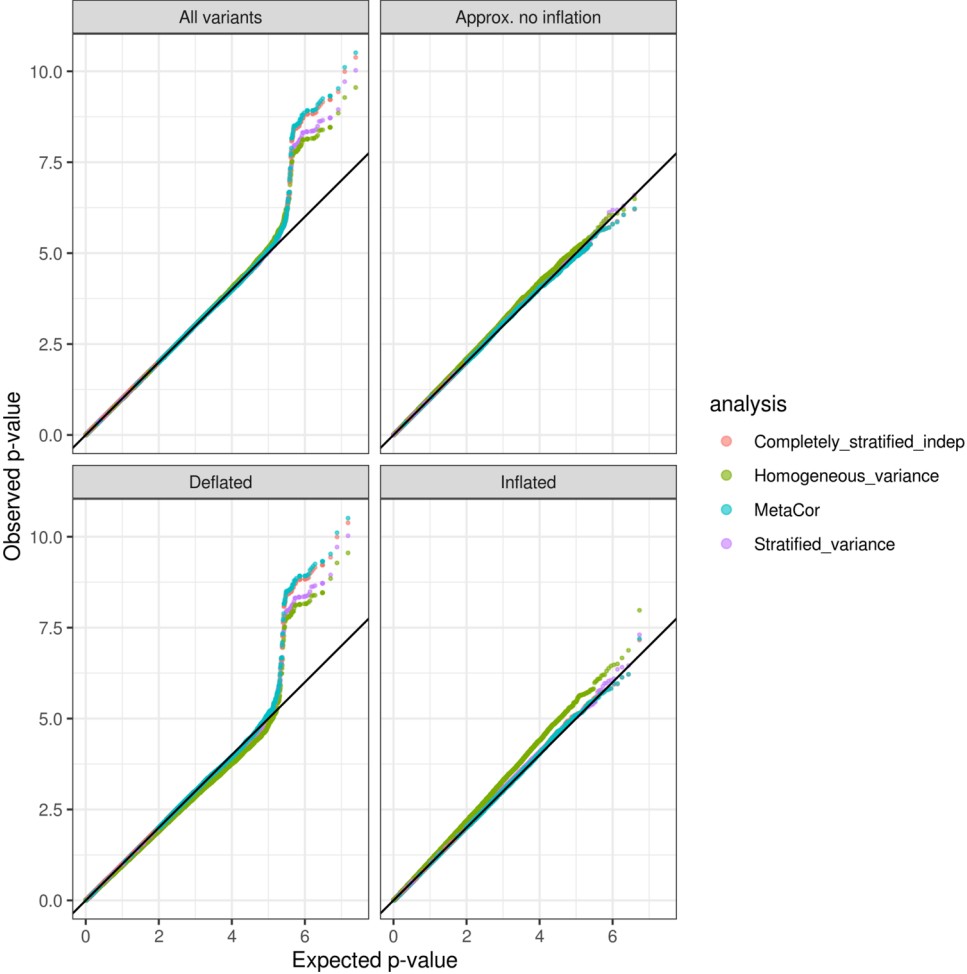

**Fig. 2 QQ-plots comparing observed and expected *p* values (−log10 transformed) from the analysis of hemoglobin concentrations.** The analyses used four approaches: "homogeneous variance" model, that assumes that all groups in the analysis have the same variances; "stratified variance" model, that allows for different residual variances across analysis groups; a "completely stratified indep" model in which analysis groups were analyzed separately, allowing for both heterogeneous residual and genetic variances across groups, and then combined together in meta-analysis under independence assumption, and "MetaCor", a procedure that accounts for relatedness across strata in the meta-analysis. The QQ-plots are provided across sets of variants classified by their inflation/deflation patterns according to the algorithm for variant-specific approximate inflation factors. We categorized variants as "Approx. no inflation" when they had estimated $\lambda_{vs}$ between 0.99 and 1.01, "Deflated" when estimated $\lambda_{vs}$ lower than 0.99, and "Inflated" when they had estimated $\lambda_{vs}$ higher than 1.01.

problem is ubiquitous for rare variants, but less so for common variants. In fact, for variants with frequency <0.05, only ~4% of variants have $\lambda_{vs}$ falling in the "approximately no inflation" category. This is because the ratio between allele frequencies has a strong effect on inflation/deflation, and ratios can become quite high when variants are rare. In the Supplementary Information, Figure S2 shows the distribution of inflation, deflation, and "approximately no inflation" categories across variants in the two analyses, and demonstrates how similar the deflation/inflation categories are between them. Most variants stay in the same category between analyses, but some rare variants (in the figure defined as MAF < 0.05) can be inflated in one analysis and deflated in the other. These differences are because $\lambda_{vs}$ coefficients are affected by sample sizes, variances, and allele frequencies, which all differ to some extent between analyses due to different samples and trait characteristics.

## Discussion

A standard tool for analysis of quantitative traits is linear or linear mixed model regression. In its widely-used default version, linear

regression is fitted under the assumption that the phenotype's residual variance is the same for all individuals in the analyzed sample. The extent of the consequences if the variances are not equal sized can be computed exactly given simplifying assumptions. Broadly, using default approaches, if a specific subgroup has a larger phenotypic variance than that of other subgroups in the pooled analysis, the estimated precision of the association signal will understate the contribution from such a subgroup. The result is deflation (loss of power) for variants where allele frequency is greater in this subgroup compared to other subgroups, or inflation (too many false positives) for variants with lower allele frequency in this subgroup compared to others.

While default linear regression methods assume the same variance for all subgroups, which leads to mis-calibrated tests if the assumption does not hold, standard computational tools can be adapted to allow for a stratified variance model, yielding better calibrated tests. Specifically, by fitting different residual variances for each study, or more generally, appropriately defined "analysis group" (e.g., all African Americans of a specific study) the problem can be alleviated. This can be viewed as fitting a different variance component for noise within each study, or as a weighted

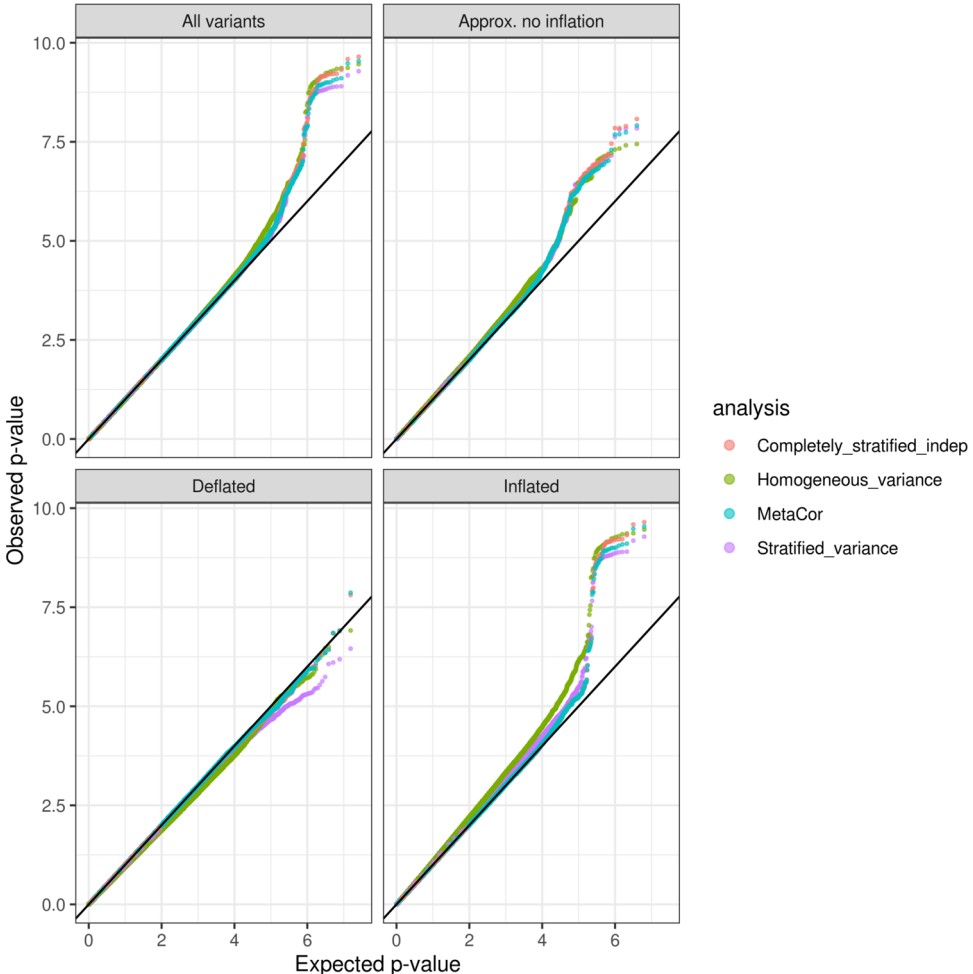

**Fig. 3 QQ-plots comparing observed and expected *p* values (−log10 transformed) from the analysis of BMI.** The analyses used four approaches: "homogeneous variance" model, which assumes that all groups in the analysis have the same variances; "stratified variance" model, which allows for different residual variances across analysis groups; a "completely stratified indep" model in which analysis groups were analyzed separately, allowing for both heterogeneous residual and genetic variances across groups, and then combined together in meta-analysis under independence assumption, and "MetaCor", a procedure that accounts for relatedness across strata in the meta-analysis. The QQ-plots are provided across sets of variants classified by their inflation/deflation patterns according to the algorithm for variant-specific approximate inflation factors. We categorized variants as "Approx. no inflation" when they had estimated $\lambda_{vs}$ between 0.99 and 1.01, "Deflated" when estimated $\lambda_{vs}$ lower than 0.99, and "Inflated" when they had estimated $\lambda_{vs}$ higher than 1.01.

least squares approach, in which the group-dependent weights are estimated. This approach is implemented in some standard genetic analysis software packages (e.g., GENESIS[9]). Our mathematical derivation and code can be used to assess the degree of miscalibration of association tests. The code uses an additive model, using a Binomial distribution for allele counts, which is commonly used in GWAS. Inflation/deflation trends should be similar between additive and dominant models, though specific values estimated using each of the two models would not be identical.

In linear regression, the stratified residual variance model allows every analysis group to have its own residual variance parameter. In the mixed model setting, where the variance is decomposed into genetic and residual variances, this model keeps the genetic variance component the same but allows for the residual variance to differ across groups. Analysis groups can be defined as study, race/ethnicity, combinations of these, or any other sample characteristics that affect trait variance and may also correlate with allele frequencies. Our mathematical derivation and code for computing $\lambda_{vs}$ are under simplifying assumptions of no covariate effects and independent observations. Therefore,

these make no distinction between genetic and residual variance components. While in the linear regression setting (independent observations) the variance stratification model clearly suffices to account for variance heterogeneity, in the mixed model setting, a residual variance stratification model may not be optimal, because it may not fully account for stratification in the genetic variance, which could be the result of study design. For example, in Fig. 5, the estimated genetic variance component of the Cleveland Family Study is much higher than those of other studies, and of the residual variance component of the same study, perhaps because study participants were selected from families with obstructive sleep apnea, which is highly associated with obesity. Heterogeneity in genetic variance is addressed in the "completely stratified model", but such a model requires that individuals are independent between different groups (strata). We also used MetaCor, a method that allows for complete stratification of analysis groups, while keeping genetically related individuals across these groups[15]. MetaCor was shown to have good statistical properties and performed well in the BMI and HGB analysis. However, it is currently computationally costlier than a pooled analysis because individual level data are used both at the

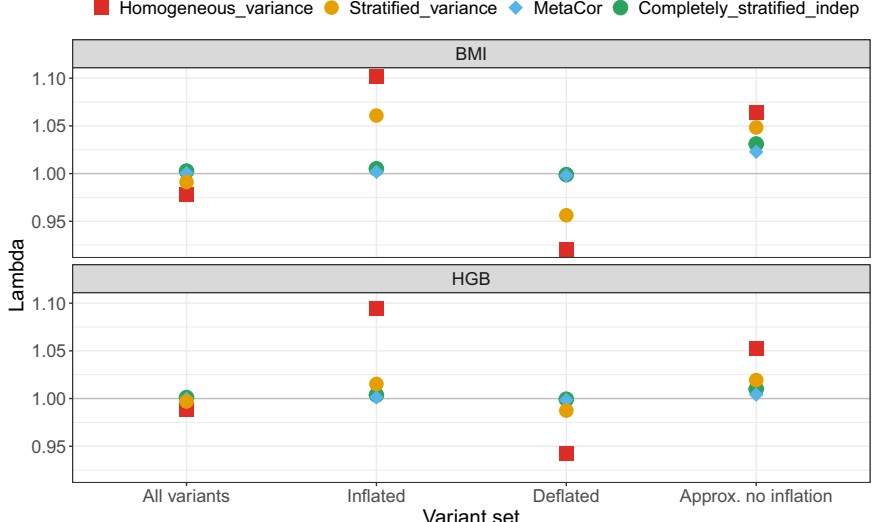

**Fig. 4 Estimated genomic control inflation factors ($\lambda_{gc}$) across compared analyses.** The figure provides estimated $\lambda_{gc}$ from the various analyzes of BMI and hemoglobin concentrations, computed across sets of variants classified by their inflation/deflation patterns according to the algorithm for approximate variant-specific inflation factors ($\lambda_{vs}$). We categorized variants as "Approx. no inflation" when they had estimated $\lambda_{vs}$ between 0.99 to 1.01, "Deflated" when estimated $\lambda_{vs}$ were lower than 0.99, and "Inflated" when they had estimated $\lambda_{vs}$ higher than 1.01. Genomic control inflation factors $\lambda_{gc}$ were computed as the ratio between the median $\chi_1^2$ test statistic across variants in the set to the theoretical median of the test statistic under the null hypothesis of no association.

**Table 1 Variant inflation/deflation characteristics by categories of MAF.**

|  |  | Deflation |  |  | Approx. no inflation | Inflation |  |  |
|---|---|---|---|---|---|---|---|---|
| MAF category | # variants | (0, 0.9] | (0.9, 0.95] | (0.95, 0.99] | (0.99, 1.01] | (1.01, 1.05] | (1.05, 1.1] | (1.1, 1.24] |
| BMI |  |  |  |  |  |  |  |  |
| [0.00102, 0.01] | 13151242 | 14.47% | 48.96% | 7.25% | 4.39% | 19.74% | 4.93% | 0.25% |
| (0.01, 0.05] | 5497411 | 0.99% | 63.77% | 6.34% | 4.14% | 24.23% | 0.53% | – |
| (0.05, 0.2] | 3393012 | – | 19.37% | 27.68% | 23.15% | 29.80% | <0.01% | – |
| (0.2, 0.5] | 3318076 | – | 0.16% | 11.40% | 70.64% | 17.80% | – | – |
| HGB |  |  |  |  |  |  |  |  |
| [0.00134, 0.01] | 11140793 | – | 64.79% | 9.05% | 3.04% | 19.97% | 3.15% | – |
| (0.01, 0.05] | 6047232 | – | 64.41% | 9.45% | 5.29% | 20.81% | 0.04% | – |
| (0.05, 0.2] | 3704075 | – | 19.59% | 29.49% | 23.54% | 27.38% | – | – |
| (0.2, 0.5] | 3358855 | – | 0.20% | 12.08% | 73.15% | 14.58% | – | – |

In each of the analyses (BMI and HGB), for each allele frequency category we provide the number of variants in this category, and from these, the proportion of variants with computed $\lambda_{vs}$ in each of multiple categories of inflation/deflation values.

individual analysis group computations, and when computing covariances between effect size estimates of all analysis groups. Computational efficiency is critical when testing the large number of variants observed in WGS studies. In addition, the MetaCor approach is not yet extended to tests of sets of rare variants (rather than single rare variants tests studied in the current manuscript). While more difficult to assess, variance stratification likely affects tests of rare variant sets as well, and methods that use a Score test based on a null model that is fit once, such as the stratified variance approach implemented in GENESIS, straightforwardly extend to such settings. As sample sizes of TOPMed grow, pooling together more diverse studies and populations, variance stratification problems may be more severe. Models allowing for pooled analysis with both group-specific residual and genetic variances or robust variance estimates may be needed for better control of Type I errors and increased efficiency. Until such methods are developed, we recommend to first use the stratified variance approach, because it is computationally efficient, it can account for relatedness across the entire sample, and the same

null model can be used to test variant sets. As a second step, we recommend computing approximate $\lambda_{vs}$, and assessing whether observed inflation/deflation remains for test statistics within groups of variants predicted to be inflated/deflated based on $\lambda_{vs}$ values. If inflation/deflation are observed despite residual variance stratification, the analyst would ideally move forward with a meta-analytic approach such as MetaCor (does not discard data but computationally more demanding), or standard meta-analysis after removing individuals to generate genetically independent strata.

## Methods

**The linear model**. For a total sample size of $n$, we assume that the data follow a linear model denoted as

$$y_i = \beta_0 + g_i\beta + \epsilon_i, \ 1 \le i \le n, \tag{1}$$

where $y_i$ is the trait or phenotype value of person $i$, $g_i$ is their count of coded alleles (i.e., genotype), $\beta_0$ denotes the mean outcome in those with no copies of the coded allele, $\beta$ denotes the effect on the mean trait of each additional copy of the coded

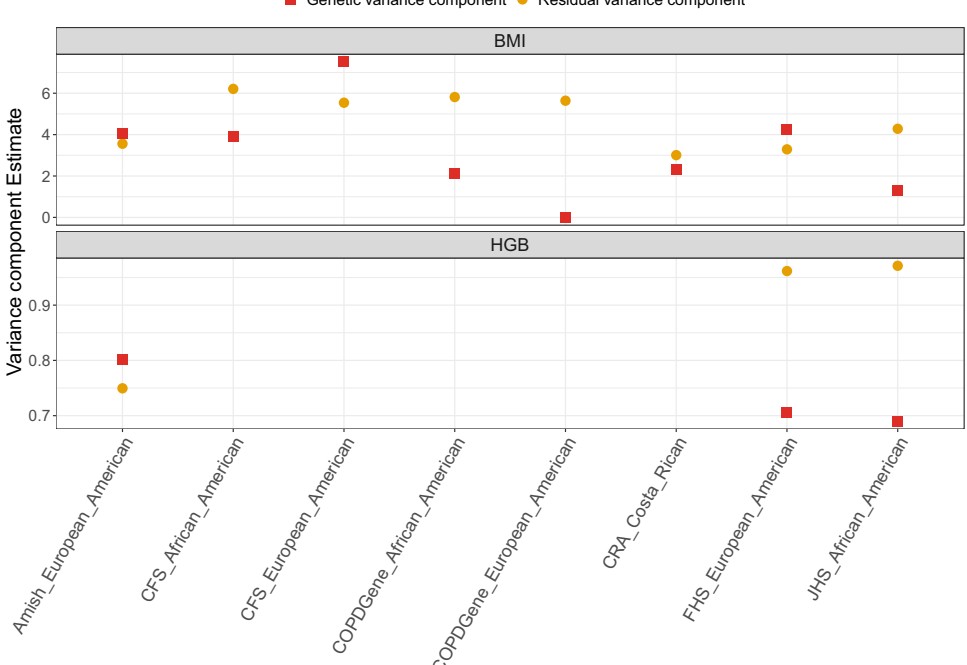

**Fig. 5 Estimated variance components across compared analyses.** The figure provides the estimated variance components corresponding to residual and genetic relatedness in the analyzes of BMI and hemoglobin concentration (HGB). For each analysis group, the estimated variance components were computed based on the analysis of the group alone, and were extracted from the second null model in the fully-adjusted two-stage rank-normalization procedure, to match the procedure used for association analysis.

allele, and the $\epsilon_i$ are residual errors, which we for now assume are independent, as a simplifying assumption.

To provide intuition for the variance stratification problem, we first demonstrate a mathematical derivation in simplified settings. We assume that the genotype effect is null ($\beta = 0$), and that the errors follow a normal distribution $\epsilon_i \sim N(0, \sigma_i^2)$. We further assume that the phenotypes are centered, the genotypes are centered, and follow a dominant mode of inheritance, i.e., we are using $g_i = (\tilde{g}_i - p)$, where $\tilde{g}_i$ is the genotype under a dominant mode (having values 1 or 0), $p$ is the frequency of having any copies of the variant allele, and $g_i$ is used in analysis.

**Implication of variance stratification on the Wald test**. The Wald test quantifies the strength of the genetic association by dividing a regression-based estimate of $\beta$ by its corresponding estimated standard error. The linear regression estimate of the effect (written in the general regression form) is

$$\hat{\beta} = \left(\sum_{i=1}^{n} g_i^2\right)^{-1} \left(\sum_{i=1}^{n} g_i y_i\right). \tag{2}$$

Denoting the estimated residual variance of individual $i$ by $\sigma_i^2$ (which may differ across individuals), the variance of $\hat{\beta}$ is

$$\text{var}(\hat{\beta}) = \left(\sum_{i=1}^{n} g_i^2\right)^{-2} \left(\sum_{i=1}^{n} g_i^2 \sigma_i^2\right). \tag{3}$$

When the variance of the residuals is homogeneous across all individuals, this is

$$\text{var}(\hat{\beta}) = \left(\sum_{i=1}^{n} g_i^2\right)^{-2} \left(\sum_{i=1}^{n} g_i^2\right)\sigma^2 = \left(\sum_{i=1}^{n} g_i^2\right)^{-1}\sigma^2, \tag{4}$$

where $\sigma^2$ is the common variance parameter. To illustrate how this approach can mislead under variance stratification, we consider the situation where two studies are present, of sizes $n_1$ and $n_2$, respectively, such that $n_1 + n_2 = n$. Further, each study is internally homogeneous with error variances $\sigma_1^2$ and $\sigma_2^2$, and it is also useful to write $p_1$ and $p_2$ for the frequency of the variant of interest (under dominant mode). Because we assume that the variant was centered in the pooled population, we have that $\text{E}[\sum_{i \in S_j} g_i^2] = n_i \text{E}[g_i^2] = n_i[p_i(1 - p_i)^2 + (1 - p_i)(0 - p_i)^2]$, or $\text{E}[\sum_{i \in S_j} g_i^2] = n_i p_i(1 - p_i)$. We can re-write Eq. (3) as:

$$\text{var}(\hat{\beta}) = \frac{n_1 p_1(1 - p_1)\sigma_1^2 + n_2 p_2(1 - p_2)\sigma_2^2}{[n_1 p_1(1 - p_1) + n_2 p_2(1 - p_2)]^2} \tag{5}$$

We see that the actual variance is a linear combination of the variance parameters

$\sigma_1^2$, $\sigma_2^2$, and the weight assigned to each depends on the minor allele frequency and sample size in each group. When the minor allele frequencies (MAFs) are equal, $p_1 = p_2$, the two forms [4] and [5] are equal, as there is no association between genotype and outcome, and no confounding occurs. But when $p_1 \neq p_2$ then the variance of the estimator upweights the residual variance in the group where the variant is more common, which does not happen under homogeneity. This result straightforwardly generalizes for $M$ studies.

In some studies, researchers use mixed models in GWAS to account for genetic relationships between individuals. Then, it is usually assumed that the variance is decomposed to an error and genetic variances, so that $\text{var}(\epsilon) = \sigma_e^2 + \sigma_g^2$. When using unrelated individuals and not accounting for genetic relatedness via a genetic relationship matrix, the two variance components are not identifiable and it is clear that accounting for differences in error variances is the same as accounting for differences in total variance (the sum of the two variance components). Musharoff et al.[8] introduced a model where the variances depends on individual-specific genetic components. For example, it could depend on a principal component (PC) of the genetic data, with: $\text{var}(\epsilon) = \sigma_e^2 + \theta_g^2 \sigma_g^2$ where $\theta_g$ is a PC, with values varying across individuals. We address this setting in simulations.

**Computing approximate variant-specific inflation factors**. We can use mathematical derivations under homogeneity and heterogeneity, relaxing the restrictive assumptions provided earlier, to compute variant-specific inflation factors. These make use of standard "sandwich" formula for large-sample approximations of the variance of estimators; for a minimally-technical summary see Result 2.1 in Wakefield[10], or for more detail Sections 5.2–5.3 of Van der Vaart[11]. We now allow for additive genetic model, and do not assume that the genotypes are standardized. The variance estimator used by the Wald test is now provided as follows:

$$\text{var}((\hat{\beta}_0, \hat{\beta})^T) = \left[\begin{pmatrix} 1 & g_1 \\ \vdots & \vdots \\ 1 & g_n \end{pmatrix}^T \begin{pmatrix} 1 & g_1 \\ \vdots & \vdots \\ 1 & g_n \end{pmatrix}\right]^{-1} \begin{pmatrix} 1 & g_1 \\ \vdots & \vdots \\ 1 & g_n \end{pmatrix}^T$$

$$\times \, cov(\mathbf{y}) \begin{pmatrix} 1 & g_1 \\ \vdots & \vdots \\ 1 & g_n \end{pmatrix} \left[\begin{pmatrix} 1 & g_1 \\ \vdots & \vdots \\ 1 & g_n \end{pmatrix}^T \begin{pmatrix} 1 & g_1 \\ \vdots & \vdots \\ 1 & g_n \end{pmatrix}\right]^{-1}$$

which simplifies if $cov(y_i) = \sigma^2$ for all $i = 1, \ldots, n$, to:

$$var((\hat{\beta}_0, \hat{\beta})^T) = \left[ \begin{pmatrix} 1 & g_1 \\ \vdots & \vdots \\ 1 & g_n \end{pmatrix}^T \begin{pmatrix} 1 & g_1 \\ \vdots & \vdots \\ 1 & g_n \end{pmatrix} \right]^{-1} \begin{pmatrix} 1 & g_1 \\ \vdots & \vdots \\ 1 & g_n \end{pmatrix}^T \sigma^2,$$

but allowing for different variances per study, it becomes:

$$var((\hat{\beta}_0, \hat{\beta})^T) = \left[ \begin{pmatrix} 1 & g_1 \\ \vdots & \vdots \\ 1 & g_n \end{pmatrix}^T \begin{pmatrix} 1 & g_1 \\ \vdots & \vdots \\ 1 & g_n \end{pmatrix} \right]^{-1} \begin{pmatrix} 1 & g_1 \\ \vdots & \vdots \\ 1 & g_n \end{pmatrix}^T$$

$$\times \begin{pmatrix} \sigma_1^2 & 0 & 0 \\ 0 & \ddots & 0 \\ 0 & 0 & \sigma_M^2 \end{pmatrix} \begin{pmatrix} 1 & g_1 \\ \vdots & \vdots \\ 1 & g_n \end{pmatrix} \left[ \begin{pmatrix} 1 & g_1 \\ \vdots & \vdots \\ 1 & g_n \end{pmatrix}^T \begin{pmatrix} 1 & g_1 \\ \vdots & \vdots \\ 1 & g_n \end{pmatrix} \right]^{-1}$$

Based on these two expressions, we propose an algorithm to compute an approximate variant-specific inflation factor. For computational purposes, we further simplify these arguments taking advantage of the fact that there are repeated rows (e.g., people who have $g_i = 1$ and are from the same study, having the same residual variance). The algorithm below uses the additional assumption that phenotypic variance within each study does not vary with genotype—which must hold under the strong null hypothesis of no association in any subpopulation. It also uses the simplifying assumption that variants are in Hardy Weinberg Equilibrium (HWE) within each study population; testing HWE is a standard preprocessing step for genotype data.

## Algorithm for computing variant-specific inflation factors

Suppose that an analyst wished to estimate a vector of regression parameters $(\beta_g, \beta_1, \ldots, \beta_M)^T$, where $\beta_g$ is a variant association measure, and $\beta_1, \ldots, \beta_M$ are intercepts for $M$ analysis groups. Denote the genotype of the $i$th individual in the $m$ analysis group by $g_{mi}$. The design matrix for estimating these parameters in linear regression would be of the form

$$\begin{pmatrix} \mathbf{g}_1 & \mathbf{1}_{n_1} & \mathbf{0}_{n_1} & \cdots & \mathbf{0}_{n_1} \\ \mathbf{g}_2 & \mathbf{0}_{n_2} & \mathbf{1}_{n_2} & \cdots & \mathbf{0}_{n_2} \\ \vdots & \vdots & \vdots & \cdots & \vdots \\ \mathbf{g}_M & \mathbf{0}_{n_M} & \mathbf{0}_{n_M} & \cdots & \mathbf{1}_{n_M} \end{pmatrix}$$

where $\mathbf{g}_m = (g_{m1}, \ldots, g_{mn_m})^T$, $1_{n_m}$ is a vector of length $n_m$ with all entries being equal to 1, and similarly $0_{n_m}$ is a vector of length $n_m$ with all entries being equal to 0. Let $V = var(y)$ be the diagonal matrix with error variances of the outcomes. The estimator of the variances and covariances of the vector of regression parameters is $var(\hat{\boldsymbol{\beta}}) = (W^T W)^{-1} W^T V W (W^T W)^{-1}$. From the matrix $var(\hat{\boldsymbol{\beta}})$ we are interested in the leading diagonal value, which is the variance of $\hat{\beta}_g$. Suppose first that one construct the matrix W using the actual data. Then:

$$(W^T W) = \begin{pmatrix} \sum_{m=1}^{M} \mathbf{g}_m^T \mathbf{g}_m & \mathbf{g}_1^T \mathbf{1}_{n_1} & \mathbf{g}_2^T \mathbf{1}_{n_2} & \cdots & \mathbf{g}_{M^T} \mathbf{1}_{n_M} \\ \mathbf{g}_1^T \mathbf{1}_{n_1} & n_1 & 0 & \cdots & 0 \\ \mathbf{g}_2^T \mathbf{1}_{n_2} & 0 & n_2 & \cdots & 0 \\ \vdots & \vdots & \vdots & \ddots & 0 \\ \mathbf{g}_{M^T} \mathbf{1}_{n_M} & 0 & 0 & 0 & n_M \end{pmatrix}$$

Now, instead of using the genotype themselves, we use the large sample limit of the expected genotype under HWE to replace the expression $\mathbf{g}_m^T 1_{n_m}$ by $n_m \times (0 \times p_m^2 + 1 \times 2p_m(1 - p_m) + 2 \times (1 - p_m)^2)$, where $n_m p_m^2$, $2n_m p_m(1 - p_m)$, and $n_m(1 - p_m)^2$ are the number of individuals from analysis group $m$ expected to have 0, 1, and 2 effect alleles under HWE. Similarly, we can replace $\mathbf{g}_m^T \mathbf{g}_m$ by its large sample limit under HWE.

Notice that the quantity $0 \times p_m^2 + 1 \times 2p_m(1 - p_m) + 2 \times (1 - p_m)^2$ is a multiplication of two vectors: $(0, 1, 2) \times (p_m^2, 2p_m(1 - p_m), (1 - p_m^T))^T$. Thus, we now define a matrix X and a matrix p, such that $(W^T W) = X^T P X$. In matrix X the left column having values $(0, 1, 2, \ldots, 0, 1, 2)^T$ $((0, 1, 2)^T$ repeating for each study), instead of the actual observed genotypes $(g_{11}, \ldots, g_{Mn_M})^T$, other columns represent study-specific intercepts, and the matrix P is a diagonal matrix providing the HWE probabilities, for each study, further scaled by the proportion of individual that each analysis group contributes to the study. We use the matrices X, P, and $V = var(y)$ to similarly replace $W^T V W$ by its large sample limit under HWE. Specifically, define:

- $X = (G\,D)$ where G is a vector of length $3M$ of the form $(0, 1, 2, \ldots, 0, 1, 2)^T$, and D is a $3M \times M$ design matrix modeling study-specific intercepts where the $i, j$ element $D_{ij}$ is

$$\begin{cases} 1 & \text{if} \quad i = 3m, 3m - 1 \, or \, 3m - 2, \, j = m \\ 0 & \text{otherwise.} \end{cases}$$

- P, a $3M \times 3M$ diagonal matrix, in which each entry gives the population proportion in each combination of genotype and study, i.e.,:

$$P = diag\left(\frac{n_1}{n} p_1^2, \frac{n_1}{n} 2p_1(1 - p_1), \frac{n_1}{n} (1 - p_1)^2, \ldots\right.$$

$$\left.\frac{n_M}{n} p_M^2, \frac{n_M}{n} 2p_M(1 - p_M), \frac{n_M}{n} (1 - p_M)^2\right)$$

V, is a $3M \times 3M$ diagonal matrix, in which each entry gives the outcome variance in each combination of genotype and study.

$$V = diag(\sigma_1^2, \sigma_1^2, \sigma_1^2, \ldots, \sigma_M^2, \sigma_M^2, \sigma_M^2)$$

Define now $B = X^T P X$ and $A = X^T P V X$, which give the large sample limits of $(W^T W)$ and $W^T V W$. Under heterogeneity the variance of the slope estimate $\hat{\beta}_g$ is proportional to the leading diagonal entry $B^{-1} A B^{-1}$. Under homogeneity the variance $\hat{\beta}_g$ is proportional to the leading entry of $B^{-1} \times sum(diag(PV))$, with the same constant of proportionality. The ratio of these two leading entries, squared, gives the large-sample value of $\lambda_{gc}$, the genomic control inflation factor[12] that would be obtained by comparing the median Wald test statistic to the median of the $\chi_1^2$ reference distribution, if all variants had the same MAF values across the studies. Because this formula provides different results for each variant, depending on the allele frequencies, we denote the ratio between the estimated values under homogeneous variance and the heterogeneous variance models $\lambda_{vs}$, for "variant specific". Note that this function requires estimation of variances (for constructing matrix P, under HWE assumption) and allele frequencies (for constructing matrix V), which are readily obtained.

An R function implementing these matrix calculations is provided, together with a tutorial that includes a coding example. These are also provided on GitHub on https://github.com/tamartsi/Variant_specific_inflation, and the function will be integrated into the GENESIS R package.

**Simulation studies.** We performed simulations to study the appropriateness of the proposed $\lambda_{vs}$, in terms of how it approximates the standard genomic control coefficient $\lambda_{gc}$ obtained from a "homogeneous variance" model that estimates a single variance parameter across data from all studies. We also studied whether a "stratified variance" model, allowing for different variance parameters across two studies, improves upon the homogeneous variances model. In this vein, we simulated unassociated genetic

variants and outcomes in a range of settings combining two studies. We simulated $n_1$, $n_2$ individuals in study 1 and study 2, $n_1 + n_2 = n$. Let $y_i$ be the outcome value of person $i$, $i = 1, \ldots, n$, and $\theta_i$ a PC value for this person, $\beta_1 = 1$, $\beta_2 = 2$ be study-specific intercepts for studies 1 and 2, $\sigma_1^2$, $\sigma_2^2$ study-specific error variances, $\sigma_g^2$ a common genetic variance parameter, and $\beta_\theta = 1$ models the linear association of the PC with the outcome. The PC was simulated from a normal distribution with variance 1, and mean $\mu_1 = 2$ in study 1, and mean $\mu_2$ in study 2 computed such that the overall PC mean in the two studies together is equal to zero (i.e., $(\sum_{i=1}^{n_2} \theta_i)/n_2 = (\sum_{j=1}^{n_1} \theta_j)/n_1$). The outcome model specified as:

$$y_i = \alpha_1 1_{study_1} + \alpha_2 1_{study_2} + \theta_i \alpha_\theta + \epsilon_i, \quad (6)$$

With

$$\epsilon_i \sim N(0, \sigma_1^2 1_{study_1} + \sigma_2^2 1_{study_2} + \sigma_g^2), \quad (7)$$

Or

$$\epsilon_i \sim N(0, \sigma_1^2 1_{study_1} + \sigma_2^2 1_{study_2} + \theta_i^2 \sigma_g^2). \quad (8)$$

In[7] the PC does not affect the genetic outcome variance, while in[8] it does. Some of the parameters were the same in all simulations (as reported above). We varied the following parameters: $n_1, n_2 \in \{1000, 5000\}$, $\sigma_1^2, \sigma_2^2, \sigma_g^2 \in \{1, 2\}$, and simulated bi-allelic independent genetic variants with MAFs $p_1, p_2 \in \{0.01, 0.05, 0.5\}$ in the two studies.

We performed 10,000 simulations for each combination of parameters and, for each such setting, computed $\lambda_{gc}$ as the ratio between median observed and expected value of the $\chi_{(1)}^2$ test statistic (under the null). We computed $\lambda_{vs}$ in each of the 10,000 simulations based estimated variances and observed allele frequencies in each of the two simulated studies, and averaged these estimates across the simulations. We compared three approaches to estimate variances[1]: fit a homogeneous variances model, obtain residuals $\hat{\epsilon}$. For each study, estimate the variance as the average $1/n \sum_{i=1}^{n} \hat{\epsilon}_i^2$ where $n$ is the number of study individuals (empirical variance)[2]; fit a "stratified variance" linear regression model allowing for different residual variances by study (as implemented in the R/Bioconductor package GENESIS[9]);[3] use the same model with stratified variances, but use the variance estimates obtained by the AI-REML algorithm (model variance). In the Supplementary Information, we provide a distribution of $\lambda_{gc}$ values that would be seen under random variability, using 10,000 independent test statistics (as was used in simulations), and simulating test statistics under the null and computing inflation factors $\lambda_{gc}$.

**Whole genome sequencing in TOPMed.** For the present analysis, we used Whole Genome Sequencing (WGS) data from freeze 4 of TOPMed. WGS was performed on DNA samples extracted from blood. Sequencing was performed by the Broad Institute of MIT and Harvard (FHS and Amish) and by the Northwest Genome Center (JHS). PCR-free libraries were constructed using commercially available kits from KAPA Biosystems (Broad) or Illumina TruSeq (NWGC). Libraries were pooled for clustering and sequencing, and later de-multiplexed using barcodes. Cluster amplification and sequencing were performed according to manufacturer's protocols using the Illumina cBot and HiSeq X sequencer, to a read depth of >30X. Base calling was performed using Illumina's Real Time Analysis 2 (RTA2) software. Read alignment, variant detection, genotype calling and variant filtering were performed by the TOPMed Informatics Research Center (University of Michigan).

Reads were aligned to the 1000 Genomes hs37d5 decoy reference sequence. Variant detection and genotype calling were performed jointly for several TOPMed studies (including the three analyzed here), using the GotCloud pipeline. Mendelian consistency was used to train a variant quality classifier using a Support Vector Machine, used for variant filtering. Additional quality control (pedigree checks, gender checks, and concordance with prior array data), performed by the TOPMed Data Coordinating Center, were used to detect and resolve sample identity issues. Further details (including software versions) are provided online (see: https://www.ncbi.nlm.nih.gov/projects/gap/cgi-bin/document.cgi?study_id=phs000964.v2.p1&phv=251960&phd=6969&pha=&pht=4838&phvf=&phdf=&phaf=&phtf=&dssp=1&consent=&temp=1).

TOPMed analyses were performed in agreement with study participants' consent, as verified via an approval process by parent studies PIs in TOPMed and TOPMed publication committee.

**Variant-specific inflation and genetic association analysis of BMI and hemoglobin concentration in TOPMed.** To demonstrate the variance stratification problem, we used datasets of hemoglobin concentrations (HGB) and body mass index (BMI) in the TOPMed freeze 4. For each of the traits, we computed a Genetic Relationship Matrix (GRM)[13] on all available variants for the corresponding trait with minor allele frequency at least 0.001, which was used to control for genetic relatedness in mixed models. Because some studies had individuals with different genetic backgrounds (leading to differences in allele frequencies), we defined "analysis groups" to use for assessment of variance stratification. An analysis group was as either all individuals from a single study (e.g., Amish), or further defined by both study and race/ethnic group (e.g., European and African Americans from the Cleveland Family Study were separate analysis groups). Thus, analysis groups capture multiple potential sources of trait variance, including differences in allele frequencies due to genetic ancestry, differences in environment and social/cultural factors, and differences in trait measurement by study. For both BMI and HGB, we performed single-variant association analysis for all variants with minor allele count of at least 20. Detailed breakdown of the studies and populations used in these analyses are provided in the Tables 2 and 3. The analysis strategy for both traits was to use the fully-adjusted two-stage procedure for rank-normalization of residuals, because it was shown to have better statistical properties (type 1 error control and power), especially when testing possibly rare genetic variants[14]. Thus, we first fit a mixed linear regression model, with fixed effects for sex, age (also age$^2$ for BMI), group defined by study and race/ethnicity, and allowing for genetic relatedness by including a variance component proportional to the GRM. Then we took the residuals generated by this model, rank-normalized them, and then re-fit the same model but with the rank-normalized residuals as the trait. For both traits, we compared four analyses: first, a 'homogeneous variance' analysis that estimates a single residual variance parameter across all individuals; second, a "stratified residual variance" model that allows a different residual variance parameter for each analysis group; third, a "completely stratified" approach which fits models and performs tests in each analysis group separately, and then combines the results via inverse-variance fixed-effects meta-analysis; and forth, a "MetaCor" analysis[15] that perform stratified analyses followed by fixed-effects meta-analysis while accounting for potential correlations due to genetic relationships between individuals in different analysis groups. The 'completely stratified' and the 'MetaCor' analyses are slightly more flexible than the stratified variance model because they allow for different genetic variance components across analysis groups, in addition to different residual variance components.

**Table 2 Analysis groups/strata participating in the BMI analysis.**

| Study | Race/ethnic group | N | Male sex number (%) | Age Mean (SD) | BMI mean (SD) |
|---|---|---|---|---|---|
| Amish | European American | 1106 | 556 (50.3%) | 51.9 (16.9) | 27.1 (4.7) |
| CFS | African American | 472 | 205 (43.4%) | 40.0 (19) | 32.7 (9.6) |
| CFS | European American | 471 | 235 (49.9%) | 44.0 (19.6) | 31.0 (9) |
| COPDGene | African American | 881 | 519 (58.9%) | 58.7 (6.7) | 28.4 (6.7) |
| COPDGene | European American | 995 | 505 (50.8%) | 63.8 (8) | 27.9 (5.7) |
| CRA | Costa Rican | 550 | 285 (51.8%) | 18.9 (16.1) | 20.8 (5.3) |
| FHS | European American | 3576 | 1584 (44.3%) | 37.2 (9) | 25.6 (4.8) |
| JHS | African American | 1756 | 641 (36.5%) | 40.3 (9.9) | 25.8 (4.7) |

For each group, we report parent study, race/ethnic group, number of participants, number and percentage of males, and age and BMI's means and standard deviations.

**Table 3 Analysis groups/strata participating in the analysis of hemoglobin concentration.**

| Study | Race/ethnic group | N | Male sex number (%) | Age mean (SD) | Hemoglobin mean (SD) |
|---|---|---|---|---|---|
| Amish | European American | 1102 | 557 (50.5%) | 50.6 (16.9) | 13.8 (1.2) |
| FHS | European American | 3133 | 1512 (48.3%) | 58.5 (15.0) | 14.1 (1.3) |
| JHS | African American | 3251 | 1212 (37.3%) | 54.8 (12.8) | 13.0 (1.5) |

For each group, we report parent study, race/ethnic group, number of participants, number and percentage of males, and age and hemoglobin's means and standard deviations.

For BMI, we removed eight individuals from the "completely stratified" analysis to ensure individuals were unrelated across groups, defined as less than third-degree relatedness. All analyses, other than MetaCor, used the GENESIS R package.

**Computing variant-specific inflation factors in mixed models with residual rank-normalization**. We studied the calibration of the various analyses of HGB and BMI by computing approximate variant-specific inflation factors $\lambda_{vs}$ and, for diagnostics, generated QQ-plots as describe later. Notably, $\lambda_{vs}$ were developed assuming independent data, and applying them in the mixed model settings provides only an approximation, as both the sample size is inaccurate (e.g., two full siblings have similar genetic data, so their effective sample size is <2), and there is more than a single variance parameter, and thus it is not straight forward to decide which variance estimates to use in computing $\lambda_{vs}$. To see that, consider the mixed- model analysis. We modeled both an error and a genetic variance component, so that for each observation, the model, in matrix form, assumes that:

$$\boldsymbol{y} = \boldsymbol{X}\beta + \boldsymbol{g_j}\alpha_j + \epsilon, \text{ with } cov(\epsilon) = \sigma_e^2\boldsymbol{I} + \sigma_g^2\boldsymbol{G}$$

Where $\boldsymbol{G}$ is the GRM, and $\sigma_e^2$, $\sigma_g^2$ are error and genetic variance components, respectively. Thus, the variance depends on $\sigma_e^2$, $\sigma_g^2$, and $\boldsymbol{G}$

In addition, we applied the fully-adjusted two-stage procedure for rank-normalization of residuals, another procedure unaccounted for by the algorithm. Therefore, different possible models will yield quite different variance estimates to be used in the $\lambda_{vs}$ computations, due to changes to the residual distributions due to rank-normalization. Because we are alerting the readers to the problems arising from assuming that variances are the same across all studies, we used variance computed based on the 'homogeneous variance' null model (the same residual and genetic variance components for all analysis groups). We extracted marginal residuals (distinguished from conditional residuals that can also be computed in mixed models) for each group, and computed empirical variance for group $j$ by $v_j = 1/n_j \sum_{i \in S_j} \widehat{\epsilon_i^2}$. Note that this estimator does not account for

relatedness. We used the residuals from the second null model from the two-stage procedure.

**Assessing population stratification at the variance level through QQ-plots**. Once $\lambda_{vs}$ are computed for each of the variants of interest, we propose to generate QQ-plots across sets of variants to visualize whether population stratification at the variance level is appropriately addressed. A function is available on the GitHub repository to generate QQ-plots stratifying variants to categories: "Inflated", "Deflated" and "Approx. no inflation". The categories can be manually defined, so that a variant can be assigned to the "Inflated" category if its $\lambda_{vs}$ is larger than a user-specified value, e.g., 1.01. Similarly, a variant is assigned to the "Deflated" category if its $\lambda_{vs}$ is lower than a user-specified, e.g., 0.99. Variants are assigned to the "Approx. no inflation" category if they are not in the "Inflated" or "Deflated" categories, i.e., their $\lambda_{vs}$ is close to the desired value of 1.

**Characterizing variants by inflation patterns**. To study how common the variant inflation/deflation problem is, and how it relates to variant frequencies, we computed the proportion of variants in levels of $\lambda_{vs}$ for each allele frequency category: <0.01, 0.01–0.05, 0.05–0.2, and 0.2–0.5. We also studied the similarity of inflation/deflation patterns between BMI and HGB.

**Reporting summary**. Further information on research design is available in the Nature Research Reporting Summary linked to this article.

**Data availability**

TOPMed (https://www.nhlbiwgs.org/) whole genome sequencing data are available, from TOPMed Freeze 5b and Freeze 8, on dbGaP by application to each of the studies used in this manuscript. Phenotypes can also be obtained through application to dbGaP. Study dbGaP accessions are: phs000956 (Amish; ncbi.nlm.nih.gov/projects/gap/cgi-bin/study. cgi?id=phs000956); phs000954 (CFS; https://www.ncbi.nlm.nih.gov/projects/gap/cgi-bin/study.cgi?id=phs000954); phs000951 (COPDGene; https://www.ncbi.nlm.nih.gov/projects/gap/cgi-bin/study.cgi?id=phs000951); phs000988 (CRA; https://www.ncbi.nlm.nih.gov/projects/gap/cgi-bin/study.cgi?study_id=phs000988.v4.p1); phs000974 (FHS; https://www.ncbi.nlm.nih.gov/projects/gap/cgi-bin/study.cgi?study_id=phs000974.v1.p1); phs000964 (JHS; https://www.ncbi.nlm.nih.gov/projects/gap/cgi-bin/study.cgi?study_id=phs000964.v4.p1). Source data are provided with this paper. Because Figs. 2 and 3 are based on results from testing tens of millions of variants, source data are provided after randomly sampling a smaller subset of data points out of those with $p$ value > 0.01. Source data are provided with this paper.

## Code availability

Statistical analyses were performed using the freely available R software, version 4.0.0. Association testing used the GENESIS package version 2.18.0, available on R/Bioconductor, or using the MetaCor R package available on GitHub https://github.com/tamartsi/MetaCor. Code for computing variant-specific inflation factors is available on GitHub https://github.com/tamartsi/Variant_specific_inflation with a tutorial, code, and example simulated data provided also in Supplementary Software 1. The code will also become available as part of GENESIS in a future release. Figures were generated using the ggplot2 R package version 3.3.0.

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

## Acknowledgements

T.S. was supported by National Heart, Lung, and Blood Institute (NHLBI; R01HL120393-03S1, 1R35HL135818, and 1R21HL145425). The views expressed in this manuscript are those of the authors and do not necessarily represent the views of the National Heart, Lung, and Blood Institute; the National Institutes of Health; or the U.S. Department of Health and Human Services. Study acknowledgements are provided in Supplementary File 1.

## Author contributions

T.S. and K.M.R. conceptualized the work and drafted the manuscript. T.S., X.Z., S.M.G., M.C., T.A.T., A.S. and K.M.R. developed, studied, and implemented the genetic analysis algorithm that incorporates different residual variances by group. T.S. performed simulation studies. X.Z., C.A.L., S.M.G. performed quality control on the genetic sequencing data. X.Z., C.A.L., J.A.B., and J.B. harmonized and performed quality control for the phenotypes used in the analysis. B.M.P., C.C.L. and K.M.R. supervised quality control and phenotype harmonization procedures. T.S., X.Z., C.A.L., S.M.G., J.A.B., M.P.C., J.C.B., T.A.T., A.S., J.R.O., E.M.L., Y.G., L.A.C., B.M.P. and K.M.R. interpreted the data, reviewed and approved the final manuscript. The NHLBI Trans-Omics for Precision Medicine (TOPMed) Consortium authors contributed to the TOPMed data collection, joint processing and quality controls, and establishment of analysis procedures.

## Competing interests

Bruce M. Psaty serves on the Steering Committee of the Yale Open Data Access Project funded by Johnson & Johnson. All other authors declare no competing interests.

## Additional information

## NHLBI Trans-Omics for Precision Medicine (TOPMed) Consortium

Namiko Abe[11], Gonçalo Abecasis[12], Francois Aguet[13], Christine Albert[14], Laura Almasy[15], Alvaro Alonso[16], Seth Ament[17], Peter Anderson[18], Pramod Anugu[19], Deborah Applebaum-Bowden[20], Kristin Ardlie[13], Dan Arking[21], Donna K. Arnett[22], Allison Ashley-Koch[23], Stella Aslibekyan[24], Tim Assimes[25], Paul Auer[26], Dimitrios Avramopoulos[21], Najib Ayas[27], Adithya Balasubramanian[28], John Barnard[29], Kathleen Barnes[30], R. Graham Barr[31], Emily Barron-Casella[21], Lucas Barwick[32], Terri Beaty[21], Gerald Beck[29], Diane Becker[21], Lewis Becker[21], Rebecca Beer[33], Amber Beitelshees[17], Emelia Benjamin[34], Takis Benos[35], Marcos Bezerra[36],

Larry Bielak[12], Joshua Bis[18], Thomas Blackwell[12], John Blangero[37], Eric Boerwinkle[38], Donald W. Bowden[39], Russell Bowler[40], Jennifer Brody[18], Ulrich Broeckel[41], Jai Broome[18], Deborah Brown[42], Karen Bunting[11], Esteban Burchard[43], Carlos Bustamante[25], Erin Buth[18], Brian Cade[44], Jonathan Cardwell[45], Vincent Carey[44], Julie Carrier[46], Cara Carty[47], Richard Casaburi[48], Juan P. Casas Romero[44], James Casella[21], Peter Castaldi[44], Mark Chaffin[13], Christy Chang[17], Yi-Cheng Chang[49], Daniel Chasman[44], Sameer Chavan[45], Bo-Juen Chen[11], Wei-Min Chen[50], Yii-Der Ida Chen[51], Michael Cho[44], Seung Hoan Choi[13], Lee-Ming Chuang[49], Mina Chung[29], Ren-Hua Chung[52], Clary Clish[13], Suzy Comhair[29], Matthew Conomos[18], Elaine Cornell[53], Adolfo Correa[54], Carolyn Crandall[48], James Crapo[40], L. Adrienne Cupples[55], Joanne Curran[37], Jeffrey Curtis[12], Brian Custer[56], Coleen Damcott[17], Dawood Darbar[57], Sean David[58], Colleen Davis[18], Michelle Daya[45], Mariza de Andrade[59], Lisa de las Fuentes[60], Paul de Vries[42], Michael DeBaun[61], Ranjan Deka[62], Dawn DeMeo[44], Scott Devine[17], Huyen Dinh[28], Harsha Doddapaneni[28], Qing Duan[63], Shannon Dugan-Perez[28], Ravi Duggirala[64], Jon Peter Durda[53], Susan K. Dutcher[60], Charles Eaton[65], Lynette Ekunwe[19], Adel El Boueiz[66], Patrick Ellinor[67], Leslie Emery[18], Serpil Erzurum[29], Charles Farber[50], Jesse Farek[28], Tasha Fingerlin[40], Matthew Flickinger[12], Myriam Fornage[38], Nora Franceschini[63], Chris Frazar[18], Mao Fu[17], Stephanie M. Fullerton[18], Lucinda Fulton[68], Stacey Gabriel[13], Weiniu Gan[33], Shanshan Gao[45], Yan Gao[19], Margery Gass[69], Heather Geiger[11], Bruce Gelb[70], Mark Geraci[35], Soren Germer[11], Robert Gerszten[71], Auyon Ghosh[44], Richard Gibbs[28], Chris Gignoux[25], Mark Gladwin[35], David Glahn[72], Stephanie Gogarten[18], Da-Wei Gong[17], Harald Goring[73], Sharon Graw[30], Kathryn J. Gray[74], Daniel Grine[45], Colin Gross[12], C. Charles Gu[68], Yue Guan[17], Xiuqing Guo[51], Namrata Gupta[13], David M. Haas[75], Jeff Haessler[69], Michael Hall[54], Yi Han[28], Patrick Hanly[76], Daniel Harris[77], Nicola L. Hawley[78], Jiang He[79], Ben Heavner[18], Susan Heckbert[18], Ryan Hernandez[43], David Herrington[39], Craig Hersh[44], Bertha Hidalgo[24], James Hixson[38], Brian Hobbs[44], John Hokanson[45], Elliott Hong[17], Karin Hoth[80], Chao (Agnes) Hsiung[52], Jianhong Hu[28], Yi-Jen Hung[81], Haley Huston[82], Chii Min Hwu[83], Marguerite Ryan Irvin[24], Rebecca Jackson[84], Deepti Jain[18], Cashell Jaquish[33], Jill Johnsen[82], Andrew Johnson[33], Craig Johnson[18], Rich Johnston[16], Kimberly Jones[21], Hyun Min Kang[12], Robert Kaplan[85], Sharon Kardia[12], Shannon Kelly[86], Eimear Kenny[70], Michael Kessler[17], Alyna Khan[18], Ziad Khan[28], Wonji Kim[66], John Kimoff[87], Greg Kinney[88], Barbara Konkle[82], Charles Kooperberg[69], Holly Kramer[89], Christoph Lange[90], Ethan Lange[45], Leslie Lange[45], Cathy Laurie[18], Cecelia Laurie[18], Meryl LeBoff[44], Sandra Lee[28], Wen-Jane Lee[83], Jonathon LeFaive[12], David Levine[18], Dan Levy[33], Joshua Lewis[17], Xiaohui Li[51], Yun Li[63], Henry Lin[51], Honghuang Lin[55], Xihong Lin[90], Simin Liu[65], Yongmei Liu[23], Yu Liu[25], Ruth J. F. Loos[70], Steven Lubitz[67], Kathryn Lunetta[55], James Luo[33], Ulysses Magalang[91], Michael Mahaney[37], Barry Make[21], Ani Manichaikul[50], Alisa Manning[92], JoAnn Manson[44], Lisa Martin[93], Melissa Marton[11], Susan Mathai[45], Rasika Mathias[21], Susanne May[18], Patrick McArdle[17], Merry-Lynn McDonald[24], Sean McFarland[66], Stephen McGarvey[65], Daniel McGoldrick[18], Caitlin McHugh[18], Becky McNeil[94], Hao Mei[19], James Meigs[67], Vipin Menon[28], Luisa Mestroni[30], Ginger Metcalf[28], Deborah A. Meyers[95], Emmanuel Mignot[96], Julie Mikulla[33], Nancy Min[19], Mollie Minear[97], Ryan L. Minster[35], Braxton D. Mitchell[17], Matt Moll[44], Zeineen Momin[28], May E. Montasser[17], Courtney Montgomery[98], Donna Muzny[28], Josyf C. Mychaleckyj[50], Girish Nadkarni[70], Rakhi Naik[21], Take Naseri[99], Pradeep Natarajan[13], Sergei Nekhai[100], Sarah C. Nelson[18], Bonnie Neltner[45], Caitlin Nessner[28], Deborah Nickerson[18], Osuji Nkechinyere[28], Kari North[63], Jeff O'Connell[17], Tim O'Connor[17], Heather Ochs-Balcom[101], Geoffrey Okwuonu[28], Allan Pack[102], David T. Paik[25], Nicholette Palmer[39], James Pankow[103], George Papanicolaou[33], Cora Parker[94], Gina Peloso[104], Juan Manuel Peralta[64], Marco Perez[25], James Perry[17], Ulrike Peters[69], Patricia Peyser[12], Lawrence S. Phillips[16], Jacob Pleiness[12], Toni Pollin[17], Wendy Post[21], Julia Powers Becker[45], Meher Preethi Boorgula[45], Michael Preuss[70], Bruce Psaty[18], Pankaj Qasba[33], Dandi Qiao[44], Zhaohui Qin[16], Nicholas Rafaels[105], Laura Raffield[63], Mahitha Rajendran[28],

Vasan S. Ramachandran[55], D. C. Rao[68], Laura Rasmussen-Torvik[106], Aakrosh Ratan[50], Susan Redline[44], Robert Reed[17], Catherine Reeves[11], Elizabeth Regan[40], Alex Reiner[107], Muagututi'a Sefuiva Reupena[108], Ken Rice[18], Stephen Rich[50], Rebecca Robillard[109], Nicolas Robine[11], Dan Roden[61], Carolina Roselli[13], Jerome Rotter[51], Ingo Ruczinski[21], Alexi Runnels[11], Pamela Russell[45], Sarah Ruuska[82], Kathleen Ryan[17], Ester Cerdeira Sabino[110], Danish Saleheen[111], Shabnam Salimi[17], Sejal Salvi[28], Steven Salzberg[21], Kevin Sandow[51], Vijay G. Sankaran[66], Jireh Santibanez[28], Karen Schwander[68], David Schwartz[45], Frank Sciurba[35], Christine Seidman[112], Jonathan Seidman[112], Frédéric Sériès[113], Vivien Sheehan[114], Stephanie L. Sherman[16], Amol Shetty[17], Aniket Shetty[45], Wayne Hui-Heng Sheu[83], M. Benjamin Shoemaker[61], Brian Silver[115], Edwin Silverman[44], Robert Skomro[116], Albert Vernon Smith[12], Jennifer Smith[12], Josh Smith[18], Nicholas Smith[18], Tanja Smith[11], Sylvia Smoller[85], Beverly Snively[39], Michael Snyder[25], Tamar Sofer[44], Nona Sotoodehnia[18], Adrienne M. Stilp[18], Garrett Storm[45], Elizabeth Streeten[17], Jessica Lasky Su[44], Yun Ju Sung[68], Jody Sylvia[44], Adam Szpiro[18], Daniel Taliun[12], Hua Tang[25], Margaret Taub[21], Kent D. Taylor[51], Matthew Taylor[30], Simeon Taylor[17], Marilyn Telen[23], Timothy A. Thornton[18], Machiko Threlkeld[18], Lesley Tinker[69], David Tirschwell[18], Sarah Tishkoff[102], Hemant Tiwari[24], Catherine Tong[18], Russell Tracy[53], Michael Tsai[103], Dhananjay Vaidya[21], David Van Den Berg[117], Peter VandeHaar[12], Scott Vrieze[103], Tarik Walker[45], Robert Wallace[80], Avram Walts[45], Fei Fei Wang[18], Heming Wang[118], Jiongming Wang[12], Karol Watson[48], Jennifer Watt[28], Daniel E. Weeks[35], Joshua Weinstock[12], Bruce Weir[18], Scott T. Weiss[44], Lu-Chen Weng[67], Jennifer Wessel[75], Cristen Willer[12], Kayleen Williams[18], L. Keoki Williams[119], Carla Wilson[44], James Wilson[120], Lara Winterkorn[11], Quenna Wong[18], Joseph Wu[25], Huichun Xu[17], Lisa Yanek[21], Ivana Yang[45], Ketian Yu[12], Seyedeh Maryam Zekavat[13], Yingze Zhang[35], Snow Xueyan Zhao[40], Wei Zhao[12], Xiaofeng Zhu[121], Michael Zody[11] & Sebastian Zoellner[12]

[11]New York Genome Center, New York, NY, USA. [12]University of Michigan, Ann Arbor, MI, USA. [13]Broad Institute, Cambridge, MA, USA. [14]Cedars Sinai, Boston, MA, USA. [15]Children's Hospital of Philadelphia, University of Pennsylvania, Philadelphia, PA, USA. [16]Emory University, Atlanta, GA, USA. [17]University of Maryland, Baltimore, MD, USA. [18]University of Washington, Seattle, WA, USA. [19]University of Mississippi, Jackson, MS, USA. [20]National Institutes of Health, Bethesda, MD, USA. [21]Johns Hopkins University, Baltimore, MD, USA. [22]University of Kentucky, Lexington, KY, USA. [23]Duke University, Durham, NC, USA. [24]University of Alabama, Birmingham, AL, USA. [25]Stanford University, Stanford, CA, USA. [26]University of Wisconsin Milwaukee, Milwaukee, WI, USA. [27]Providence Health Care, Vancouver, BC, Canada. [28]Baylor College of Medicine Human Genome Sequencing Center, Houston, TX, USA. [29]Cleveland Clinic, Cleveland, OH, USA. [30]University of Colorado Anschutz Medical Campus, Aurora, Colorado, USA. [31]Columbia University, New York, NY, USA. [32]The Emmes Corporation, Rockville, MD, USA. [33]National Heart, Lung, and Blood Institute, National Institutes of Health, Bethesda, MD, USA. [34]Boston University, MA General Hospital, Boston, MA, USA. [35]University of Pittsburgh, Pittsburgh, PA, USA. [36]Fundaçáo de Hematologia e Hemoterapia de Pernambuco—Hemope, Recife, BR, Brazil. [37]University of Texas Rio Grande Valley School of Medicine, Brownsville, TX, USA. [38]University of Texas Health at Houston, Houston, TX, USA. [39]Wake Forest Baptist Health, Winston-Salem, NC, USA. [40]National Jewish Health, Denver, CO, USA. [41]Medical College of Wisconsin, Milwaukee, WI, USA. [42]University of Texas Health at Houston, Houston, TX, USA. [43]University of California, San Francisco, San Francisco, CA, USA. [44]Brigham & Women's Hospital, Boston, MA, USA. [45]University of Colorado at Denver, Denver, CO, USA. [46]University of Montreal, Montreal, WI, USA. [47]Washington State University, Pullman, WA, US. [48]University of California, Los Angeles, Los Angeles, CA, USA. [49]National Taiwan University, Taipei, Taiwan, ROC. [50]University of Virginia, Charlottesville, VA, USA. [51]Lundquist Institute, Torrance, CA, USA. [52]National Health Research Institute, Miaoli County, Taiwan, ROC. [53]University of Vermont, Burlington, VT, USA. [54]University of Mississippi, Jackson, MS, USA. [55]Boston University, Boston, MA, USA. [56]Vitalant Research Institute, San Francisco, CA, USA. [57]University of Illinois at Chicago, Chicago, IL, USA. [58]University of Chicago, Chicago, IL, USA. [59]Mayo Clinic, Rochester, MN, USA. [60]Washington University in St Louis, St. Louis, MO, USA. [61]Vanderbilt University, Nashville, TN, USA. [62]University of Cincinnati, Cincinnati, OH, USA. [63]University of North Carolina, Chapel Hill, NC, USA. [64]University of Texas Rio Grande Valley School of Medicine, Edinburg, TX, USA. [65]Brown University, Providence, RI, USA. [66]Harvard University, Cambridge, MA, USA. [67]Massachusetts General Hospital, Boston, MA, USA. [68]Washington University in St Louis, St Louis, MO, USA. [69]Fred Hutchinson Cancer Research Center, Seattle, WA, USA. [70]Icahn School of Medicine at Mount Sinai, New York, NY, USA. [71]Beth Israel Deaconess Medical Center, Boston, MA, USA. [72]Boston Children's Hospital, Harvard Medical School, Boston, MA, USA. [73]University of Texas Rio Grande Valley School of Medicine, San Antonio, TX, USA. [74]Mass General Brigham, Boston, MA, USA. [75]Indiana University, Indianapolis, IN, USA. [76]University of Calgary, Calgary, AB, Canada. [77]University of Maryland, Philadelphia, PA, USA. [78]Yale University, New Haven, CT, USA. [79]Tulane University, New Orleans, LO, USA. [80]University of Iowa, Iowa City, IA, USA. [81]Tri-Service General Hospital National Defense Medical Center, Taipei City, Taiwan, ROC. [82]Blood Works Northwest, Seattle, WA, USA. [83]Taichung Veterans General Hospital Taiwan, Taichung City, Taiwan, ROC. [84]Oklahoma State University Medical Center, Columbus, OH, USA. [85]Albert Einstein College of Medicine, New York, NY, USA. [86]University of California, San Francisco, San Francisco, CA, USA. [87]McGill University, Montreal, QC, Canada. [88]University of Colorado at Denver, Aurora, CO, USA. [89]Loyola University, Maywood, IL, USA. [90]Harvard School of Public Health, Boston, MA, USA. [91]Ohio State University, Columbus, OH, USA. [92]Broad Institute, Harvard University, Massachusetts General Hospital, Boston, MA, USA. [93]George Washington University, Washington, DC, USA. [94]RTI International, Research Triangle Park, USA. [95]University of Arizona, Tucson, AZ, USA. [96]Stanford University, Palo Alto, California, USA. [97]National Institute of Child Health and Human Development, National Institutes of Health, Bethesda, MD, USA. [98]Oklahoma Medical Research Foundation, Oklahoma City, OK, USA. [99]Ministry

of Health, Government of Samoa, Apia, WS, Samoa. [100]Howard University, Washington, DC, USA. [101]University at Buffalo, Buffalo, NY, USA. [102]University of Pennsylvania, Philadelphia, PA, USA. [103]University of Minnesota, Minneapolis, MN, USA. [104]Boston University, Boston, MA, USA. [105]University of Colorado at Denver, Denver, CO, USA. [106]Northwestern University, Chicago, IL, USA. [107]Fred Hutchinson Cancer Research Center, University of Washington, Seattle, WA, USA. [108]Lutia I Puava Ae Mapu I Fagalele, Apia, WS, Samoa. [109]University of Ottawa, Ottawa, ON, Canada . [110]Universidade de Sao Paulo, Sao Paulo, Brazil. [111]Columbia University, New York, New York, USA. [112]Harvard Medical School, Boston, MA 02115, USA. [113]Université Laval, Quebec City, QC, Canada. [114]Emory University, Atlanta, GA, USA. [115]UMass Memorial Medical Center, Worcester, MA, USA. [116]University of Saskatchewan, Saskatoon, SK, Canada. [117]University of Southern California, Los Angeles, CA, USA. [118]Brigham & Women's Hospital, Mass General Brigham, Boston, MA, USA. [119]Henry Ford Health System, Detroit, MI, USA. [120]Beth Israel Deaconess Medical Center, Cambridge, MA, USA. [121]Case Western Reserve University, Cleveland, OH, USA.

