## [Peer Review File · Nature Communications]

REVIEWER COMMENTS

Reviewer #2 (Remarks to the Author):

Sofer et al. present a method and an associated software implementation (R code) to better account for heterogeneity in phenotype variability between study populations in pooled whole-genome association studies. They propose to calculate variant-specific inflation factors by stratifying the samples into more homogenous groups. They present a simulation study and comparisons with existing approaches, also using real data from TopMED. While the figures show somewhat better control as compared to the standardly applied method assuming homogeneous variance across individuals, there are some concerns that I detail below.

1. In the introduction, the authors point out to three previous studies (6-8) that addressed the same issue. It is unclear whether the proposed methods have some disadvantages and/or what is the advantage of the currently proposed method over the previously proposed one. It also seems from the Methods that the authors have built their proposed work on the method proposed by Musharoff et al (ref 8). Could the authors clarify on the novelty of their proposed method and the comparison with the existing ones?

2. The authors discuss the effect of different MAFs in different populations to the variance of the estimator. It is clear from the formula that also unequal sample sizes will have an effect on the weight of the variances and thus, variance of the estimator. It seems that the effect of different sample sizes was also tested in the simulation setting but the results are not shown. Could the authors comment on this?

3. The effects of different MAFs are not shown. This is even more relevant because the authors show good control with already established MetaCor but discuss that it is only introduced for the Wald test but not for the Score test, often used for the rare variants. How does the proposed method work with rare variants? The authors do point out the computational cost of MetaCor as a drawback, in addition to the rare variant issue, but show no solution from their method to the application of rare variant analysis.

4. How about comparison with GENESIS that the authors mention in the discussion?

5. How does the method perform under various underlying genetic architectures, e.g. additive, dominant and recessive models?

6. In the QQ plots it is very hard to see any differences between the different approaches, except for some clear deviations, such as the left bottom panel in Fig. 3 that the authors also point out. The authors could try to use a smaller dot size to distinguish the lines. It would be also good if the authors could provide some numerical results in addition to the figures. Also, some numerical results from the haemoglobin and BMI analyses would be useful. Are they detecting e.g. more false-positive signals in the standard approach analyses vs the proposed one? How many genome-wide significant signals are detected from each approach? What would be the conclusions of a WGS on these phenotypes from the different approaches?

7. "About right" category might be better named as "No inflation".

Reviewer #4 (Remarks to the Author):

The manuscript from Sofer et al describes a novel approach to assessing the variability of variant-level parameters across stratified cohorts when performing pooled genetic analysis studies, particularly as in WGS. The authors compare two models of variance stratification (homogenous or stratified) under a variety of simulated scenarios to determine that allowing for differential variance between studies using the stratified model maintained an appropriate genomic inflation factor across a range of variance inflation models. They also demonstrate the problem using actual TOPMed sequencing data from several diverse cohort. The approach is novel, interesting, and

relevant.

1. It would be helpful to the casual reader to have a bit more description on the expected/desired results in Figure 1. Also given the very tight estimates for the low-variance scenarios it might be better to allow a change in axes to view any deviation in the data points from 0,0.
2. How were the "deflated", "inflated" and "about right" bin cutoff points chosen?
3. How many variants in total were evaluated in the TOPMed analyses? How many variants fall into each bin for the different analyses? Are rarer SNPs likely to have more variance stratification due to numerically subtle MAF differences which are proportionally large with lower frequency? It would be helpful to see some characteristics of the variant bins for each analysis, at least counts and some frequency metrics (proportions below some threshold seem possibly more useful than mean frequency).
4. It is interesting that the Q-Q plots show inflation of the "All" and "Deflated" groups for HGB and for all groups except "Deflated" for BMI. I suppose this may illustrate the differential nature of variance stratification across phenotypes and cohort composition. It would be interesting to note proportions of variants which are consistently in i.e. the deflated bin across the two exemplar studies compared to those which change bins, as the underlying cause of the variance stratification may be changing with phenotype or group composition.
5. It is unclear what the author's recommendation is at the conclusion of this study. It seemed that stratification by study followed by meta-analysis performed quite well, and metaCor which also performed well and was robust to variance stratification is applicable only in the computationally expensive Wald test. A concluding statement defining what the authors feel are "best practices" or preferred approaches/solutions to addressing this challenging scenario would be nice to see.

Reviewer #5 (Remarks to the Author):

The authors address the problem of phenotype stratification when pooling studies of different origins. While the impact of genetic heterogeneity has been extensively addressed, this is less the case for the phenotypic part.

Explicit derivation of beta for a two-populations models helps states the problem. It is clear, as already shown, that there is inflation when there is allele frequency difference.

Authors propose a robust sandwich variance estimator to better correct for the true inflation.

Moreover, they derive allele specific (or MAF specific) inflation factor to better calibrate the tests.

The method is tested on simulations and on real data (TopMed). Inflation factors are estimated for various methods, either "naïve" to variance stratification or some specifically designed like MetaCor.

Comments :

1/ I think that performing meta-analysis is more common than expected. The authors state that with WGS data we tend to pool data rather than performing meta-analyses like in GWAS chip data. My feeling is that it is not the nature of the data but maybe the fact that we run rare variants tests, which do not lend themselves as much to meta-analysis as frequent variant which is the cause of pooling. Even though methods for rare variant burden test are well developed now. The authors is still important but maybe they could present data pooling rather as a more powerful approach – especially for rare variants.

2/ Description of the algorithm : the authors should present more clearly the parameters in their algorithm. D, G, M are not previously defined although we understand in passing that this is the number of groups.

2 bis/ I presume that matrix P, using proportion of genotypes within each sub-population is key to the simplification algorithm (factoring on individuals with same genotype within same population). However, for less abstract minds, like the one of your reviewer, I would propose a graph that explains how the genotypes are embedded in the vector/matrix.

3/ Simulations : the authors are simulating a difference in Principal Component in order to mimic population genetic heterogeneity. Would it be possible to know which kind of heterogeneity is simulated here – like is it intercontinental for instance ... which Fst could this correspond to ...

4/ While the discussion recalls some of the properties of the linear model that might be more

relevant in introduction, it fails, for me, in giving a kind of advice on how to proceed. MetaCor looks about right but is too costly, so what are the authors recommendations ? I guess this has something to do with running the most robust of the three methods (excluding MetaCor) and apply the proposed inflation correction ? This should be made clear as this is, according to me, what this paper is about.

Response to review of NCOMMS-20-09619A-Z “Variant-Specific Inflation Factors for Assessing Population Stratification at the Phenotypic Variance Level”

We thank the editor and the three reviewers for considering our manuscript and for their thoughtful reviews. We provide item-by-item responses below.

Reviewer #2:

Sofer et al. present a method and an associated software implementation (R code) to better account for heterogeneity in phenotype variability between study populations in pooled whole-genome association studies. They propose to calculate variant-specific inflation factors by stratifying the samples into more homogenous groups. They present a simulation study and comparisons with existing approaches, also using real data from TopMED. While the figures show somewhat better control as compared to the standardly applied method assuming homogeneous variance across individuals, there are some concerns that I detail below.

Response: thank you for your thorough review and excellent comments.

1. In the introduction, the authors point out to three previous studies (6-8) that addressed the same issue. It is unclear whether the proposed methods have some disadvantages and/or what is the advantage of the currently proposed method over the previously proposed one. It also seems from the Methods that the authors have built their proposed work on the method proposed by Musharoff et al (ref 8). Could the authors clarify on the novelty of their proposed method and the comparison with the existing ones?

Response: Thank you for this comment. To clarify, the contribution of our manuscript is the development of the variant-specific inflation factors, which we use to demonstrate inflation and deflation caused by variance heterogeneity coupled with allele frequency differences. The other work cited records variance heterogeneity and studies its effect on association testing,

but has no element of variant-by-variant diagnoses of this problem. To clarify this, we edited the last paragraph of the introduction (page 4, bottom), which now reads (new text in italics):

“In this manuscript, we develop variant-specific inflation factors λ_{vS} , which quantify the degree of inflation/deflation in association testing of a single genetic variant due to population stratification at the variance level. We develop an algorithm to compute approximate variant-specific inflation factors based on allele frequencies and variances in groups pooled together for analysis, demonstrate their usage for assessing model fit, and demonstrate the implications of the population stratification at the variance level in simulations and in analyses of WGS data from TOPMed. *To account for population stratification at the variance level we use the computationally efficient and scalable approach proposed in Conomos et al (7) and implemented in GENESIS (9), and show in simulations that it indeed addresses the variance stratification problem in scenarios based on Musharoff et al (8).*”

2. The authors discuss the effect of different MAFs in different populations to the variance of the estimator. It is clear from the formula that also unequal sample sizes will have an effect on the weight of the variances and thus, variance of the estimator. It seems that the effect of different sample sizes was also tested in the simulation setting but the results are not shown. Could the authors comment on this?

Response: We did use unequal sample sizes as well but have chosen to not present this detail, because with many potential combinations of parameters it becomes unwieldy. Instead, Figure 1 provides simulation results in a grouped manner. Each “simulation setting” correspond to multiple simulations in which a few characteristics are satisfied, but within them we have distributions of sample sizes, specific allele frequencies, etc. To address your comment, we added information in the figure legend, which now reads (new text in italics):

“The figure compares estimated variant-specific inflation factors λ_{vs} estimated in each of many simulation settings, and corresponding observed inflation λ_{gc} based on 10,000 repetitions of each simulation settings. Observed inflation values are provided based on a homogeneous variance model, in which a single variance parameter is estimated using the aggregated data; and based on a stratified variance model, that fits a different variance parameter to each of the two simulated studies. Each simulation setting corresponds to a single point on this figure, and the simulations are grouped (denoted by different colors and symbols) by the characteristics stated in the legend. Within each group of simulation settings, the simulation parameters differ by specific parameter values, including MAFs, variance components, and sample sizes, while still satisfying the broad conditions of the grouped simulation settings. Estimates of variant-specific inflation factors were averaged across 10,000 simulation repetitions. The dashed horizontal lines corresponds to the 2.5% and 97.% quantiles of the distribution of λ_{gc} based on 10,000 variants under the null of no inflation/deflation, obtained from simulations. “

3. The effects of different MAFs are not shown. This is even more relevant because the authors show good control with already established MetaCor but discuss that it is only introduced for the Wald test but not for the Score test, often used for the rare variants. How does the proposed method work with rare variants? The authors do point out the computational cost of MetaCor as a drawback, in addition to the rare variant issue, but show no solution from their method to the application of rare variant analysis.

Response: we apologize for the confusion. Our proposed diagnostic method is for computing variant-specific inflation factors. For this, we did consider scenarios of differences in MAFs across combined studies, and a range of frequencies (simulation studied visualized in Figure 1). In data analysis, we also considered rare variants: we tested variants with minor allele count of at least 20. Therefore, for testing single rare variants, the conclusions apply, and we deleted the statement about the Wald test when discussing MetaCor (because the limitation is of computational, rather than statistical, properties).

We made a few revisions in the text to clarify what is our contribution (development of the variant-specific inflation factors), and the methods that we used (GENESIS R package), as explained in response to your earlier comments. In addition, your comment prompted us to refer to another limitation of MetaCor: the (lack of) application for testing sets of rare variants. We address this in the discussion (page 23, bottom), as follows: *“In addition, the MetaCor approach is not yet extended to tests of sets of rare variants (rather than single rare variants tests studied in the current manuscript). While more difficult to assess, variance stratification likely affects tests of rare variant sets as well, and methods that use a Score test based on a null model that is fit once, such as the stratified variance approach implemented in GENESIS, straightforwardly extend to such settings.”*

4. How about comparison with GENESIS that the authors mention in the discussion?

Response: We apologize for the confusion, we clarified that we did use GENESIS in our analyses (other than MetaCor). At the end of the Methods section *“Variant-specific inflation and genetic association analysis of BMI and hemoglobin concentration in TOPMed”* (page 16, top), we added the statement *“All analyses, other than MetaCor, used the GENESIS R package.”*.

5. How does the method perform under various underlying genetic architectures, e.g. additive, dominant and recessive models?

Response: We compute variant-specific inflation factors based on an additive model, which is what implemented in major genetic analysis software and customary to use in GWAS. The extension for dominant and recessive mode requires relying on the Bernoulli distribution rather than the Binomial distribution in the computation, with the parameter of the distribution being the frequency of the allele combination of interest (having at least one effect allele in a dominant mode, or two effect alleles in a recessive mode). Since our mathematical derivation was performed under a dominant/recessive mode (i.e., using the Bernoulli distribution), we indeed expect that the trends (inflation/deflation) are the same between the two approaches,

despite the specific values of the variant-specific inflation factors being somewhat different. To address your comment, we added the following to the second paragraph of the discussion: “Our mathematical derivation and code can be used to assess the degree of miscalibration of association tests. For exposition, the mathematical derivation uses the simplifying assumption of two groups, and recessive or dominant model, where alleles are sampled from a Bernoulli distribution. The code uses an additive model, using Binomial distribution for allele counts, which is commonly used in GWAS. Inflation/deflation trends should be similar between the two models, though specific values estimated using dominant and additive models would be not be identical.”.

6. In the QQ plots it is very hard to see any differences between the different approaches, except for some clear deviations, such as the left bottom panel in Fig. 3 that the authors also point out. The authors could try to use a smaller dot size to distinguish the lines. It would be also good if the authors could provide some numerical results in addition to the figures. Also, some numerical results from the haemoglobin and BMI analyses would be useful. Are they detecting e.g. more false-positive signals in the standard approach analyses vs the proposed one? How many genome-wide significant signals are detected from each approach? What would be the conclusions of a WGS on these phenotypes from the different approaches?

Response: Thank you for these helpful suggestions. We have reduced the point size. Regarding the number of genome-wide significant SNPs, this is not the focus of our work, which is on calibration of tests over broad classes of SNPs. Moreover, interpreting differences in significance between methods is challenging: with no gold standard available for the truth at any SNP, and very likely modest power for a great majority of true signals, distinguishing false and true positive will be unreliable for any individual result, or any small set of results as suggested here.

Regarding the WGS results, the work we report here could supplement results from standard methods (homogeneous variance Wald or Score tests) with indications of which SNPs had

sufficiently divergent MAFs, depending on study-specific differences in phenotypic variance, that non-constant variance by study should be considered a plausible explanation for putative signals.

7. “About right” category might be better named as “No inflation”.

Response: Thank you, we agree and now use “Approximately no inflation” for clarity.

Reviewer #4 (Remarks to the Author):

The manuscript from Sofer et al describes a novel approach to assessing the variability of variant-level parameters across stratified cohorts when performing pooled genetic analysis studies, particularly as in WGS. The authors compare two models of variance stratification (homogenous or stratified) under a variety of simulated scenarios to determine that allowing for differential variance between studies using the stratified model maintained an appropriate genomic inflation factor across a range of variance inflation models. They also demonstrate the problem using actual TOPMed sequencing data from several diverse cohort. The approach is novel, interesting, and relevant.

Response: Thank you for a thoughtful and helpful review!

1. It would be helpful to the casual reader to have a bit more description on the expected/desired results in Figure 1. Also given the very tight estimates for the low-variance scenarios it might be better to allow a change in axes to view any deviation in the data points from 0,0.

Response: We agree and have updated the figure to have different scales between groups of settings (rows of panels in the figure).

In making this change, we found it helpful to show that inflation/deflation is controlled when allele frequencies are the same and/or variances are the same between populations, and the noise shown on Fig 1 (bottom panels) is just the small number of simulations (10,000 in each settings). In the supplementary material we now explain this and use a small simulation study to demonstrate it.

We made the following revisions in the text. First, at the end of the description of the simulations in the Methods section (subsection “*Simulation studies*”, page 13, middle) we added the text: “We provide a reference to potential values of λ_{gc} that can plausibly be estimated when using 10,000 independent test statistics (as was used in simulations), by simulating test statistics under the null and computing inflation factors λ_{gc} . This is described in the Supplementary Materials.”.

Next, we re-wrote the description of the results visualized in Figure 1, in the Results section, subsection “*Simulation studies*” (pages 18-19). This is the updated text, new parts in italics:

“The top three rows in Figure 1 demonstrate settings in which both the MAF and the total variances differ between the two combined studies, including settings in which both the error and genetic variance components are the same, but the PC affects the genetic variance, resulting in different total variance between the studies because the mean of the PC differ between them. In these settings the variance stratification is observed when using the homogeneous variance model, in that the observed inflation can be substantially higher or lower than 1, which exact values depending on the specific parameters used in each simulation. Indeed, the computed λ_{vs} and the observed λ_{gc} are highly correlated. In contrast, the stratified variance model was robust to variance stratification across all settings, with observed inflation around 1 in all simulations. The bottom two rows of Figure 1 demonstrate settings in which

either the MAF or the variances are the same in the two combined studies. In these settings the expected inflation computed by λ_{vS} is always 1 (no inflation). As expected, the observed inflation is the same in the homogeneous and stratified variance models. The spread seen in the values of the observed inflation, with some values higher and some lower than the desired 1, are consistent with that expected based on the number of replication of simulations in each setting (10,000); see Supplementary Materials for more details.”

2. How were the “deflated”, “inflated” and “about right” bin cutoff points chosen?

Response: These choices are necessarily somewhat arbitrary. Our choices ($\lambda=0.99$ and 1.01) were intended to give a central range where most researchers would typically be reassured by the results, as any mis-statement in statistical significance is likely to be minor. With less skeptical researchers, one might consider a wider range of λ values for the “about right” category, but the general form of the results would stay the same.

3. How many variants in total were evaluated in the TOPMed analyses? How many variants fall into each bin for the different analyses? Are rarer SNPs likely to have more variance stratification due to numerically subtle MAF differences which are proportionally large with lower frequency? It would be helpful to see some characteristics of the variant bins for each analysis, at least counts and some frequency metrics (proportions below some threshold seem possibly more useful than mean frequency).

Response: Thank you for these excellent suggestions. We made the following revisions in response. First, at the end of the Methods section (page 17, bottom), we added a subsection: *“Characterizing variants by inflation patterns*

To study how common the variant inflation/deflation problem is, and how it relates to variant frequencies, we computed the proportion of variants in levels of λ_{vS} for each allele frequency

category: <0.01, 0.01-0.05, 0.05-0.2, and 0.2-0.5. We also studied how similar the inflation/deflation patterns between BMI and HGB.”

Next, at the end of the Results, we added a paragraph referring to this and your comment 4. In response to this comment, we wrote (page 20, bottom):

“Table 3 describes the inflation/deflation patterns of variants according to their MAF. One can see that the inflation/deflation problem is ubiquitous for rare variants, but less so for common variants. In fact, for variants with frequency < 0.05, only ~4% of variants have λ_{vs} falling in the “approximately no inflation” category. This is because the ratio between allele frequencies has strong effect on inflation/deflation, and ratios can become quite high when variants are rare.”

Finally, we added Table 3. For convenience of review, it is copied here:

Table 3: Variant inflation/deflation characteristics by categories of MAF.

MAF category	# variants	Deflation			Approx. no inflation (0.99,1.01]	Inflation		
		(0,0.9]	(0.9,0.95]	(0.95,0.99]		(1.05,1.1]	(1.1,1.24]	(1.24,1.5]
BMI								
[0.00102,0.01]	1315124	14.47%	48.96%	7.25%	4.39%	19.74%	4.93%	0.25%
(0.01,0.05]	5497411	0.99%	63.77%	6.34%	4.14%	24.23%	0.53%	--
(0.05,0.2]	3393012	--	19.37%	27.68%	23.15%	29.80%	<0.01%	--
(0.2,0.5]	3318076	--	0.16%	11.40%	70.64%	17.80%	--	--
HGB								
[0.00134,0.01]	1114079	--	64.79%	9.05%	3.04%	19.97%	3.15%	--
(0.01,0.05]	6047232	--	64.41%	9.45%	5.29%	20.81%	0.04%	--
(0.05,0.2]	3704075	--	19.59%	29.49%	23.54%	27.38%	--	--
(0.2,0.5]	3358855	--	0.20%	12.08%	73.15%	14.58%	--	--

In each of the analyses (BMI and HGB), for each allele frequency category we provide the number of variants in this category, and from these, the proportion of variants with computed λ_{vs} in each of multiple categories of inflation/deflation values.

4. It is interesting that the Q-Q plots show inflation of the “All” and “Deflated” groups for HGB and for all groups except “Deflated” for BMI. I suppose this may illustrate the differential nature of variance stratification across phenotypes and cohort composition. It would be interesting to note proportions of variants which are consistently in i.e. the deflated bin across the two exemplar studies compared to those which change bins, as the underlying cause of the variance stratification may be changing with phenotype or group composition.

Response: as we alluded to in the response to your comment 3, we performed such an analysis. Interestingly, most variants stayed in the same category. We added the following text in the Results section (page 20, bottom): “In the Supplementary Materials, Figure S2 visualizes the distribution of inflation, deflation, and “approximately no inflation” categories across variants in the two analyses, and demonstrates how similar the deflation/inflation categories are between them. Most variants stay in the same category between analyses, but some rare variants (in the figure defined as $MAF < 0.05$) can be inflated in one analysis and deflated in the other. These differences are because λ_{vS} are affected by sample sizes, variances, and allele frequencies, all differ to some extent between analyses due to different samples and trait characteristics.”, and in the supplementary materials, Figure S2, which we put here for convenience:

Figure S2: Distribution and overlap in inflation/deflation categories of variants used in the HGB and BMI analyses.

The Y-axis provides the number of variants counted, in thousands.

If you think this should be in the main manuscript rather than the supplementary materials, we will be happy to move it there.

5. It is unclear what the author's recommendation is at the conclusion of this study. It seemed that stratification by study followed by meta-analysis performed quite well, and metaCor which also performed well and was robust to variance stratification is applicable only in the computationally expensive Wald test. A concluding statement defining what the authors feel are "best practices" or preferred approaches/solutions to addressing this challenging scenario would be nice to see.

Response: We have now added the following sentences at the end of the discussion (page 23, bottom): *"Until such methods are developed, we recommend to first use the stratified variance approach, because it is computationally efficient, it can account for relatedness across the entire sample, and the same null model can be used to test variant sets. As a second step, we*

recommend computing approximate λ_{vS} , and assessing whether observed inflation/deflation remains for test statistics within groups of variants predicted to be inflated/deflated based on λ_{vS} values. If inflation/deflation are observed despite residual variance stratification, the analyst would ideally move forward with a meta-analytic approach such as MetaCor (does not discard data but computationally more demanding), or standard meta-analysis after removing individuals to generate genetically independent strata.”.

Reviewer #5 (Remarks to the Author):

The authors address the problem of phenotypic stratification when pooling studies of different origins. While the impact of genetic heterogeneity has been extensively addressed, this is less the case for the phenotypic part.

Explicit derivation of beta for a two-populations models helps states the problem. It is clear, as already shown, that there is inflation when there is allele frequency difference.

Authors propose a robust sandwich variance estimator to better correct for the true inflation. Moreover, they derive allele specific (or MAF specific) inflation factor to better calibrate the tests. The method is tested on simulations and on real data (TopMed). Inflation factors are estimated for various methods, either “naïve” to variance stratification or some specifically designed like MetaCor.

Response: thank you for your review.

Comments :

1/ I think that performing meta-analysis is more common than expected. The authors state that with WGS data we tend to pool data rather than performing meta-analyses like in GWAS chip data. My feeling is that it is not the nature of the data but maybe the fact that we run rare variants tests, which do not lend themselves as much to meta-analysis as frequent variant which is the cause of pooling. Even though methods for rare variant burden test are well

developed now. The authors is still important but maybe they could present data pooling rather as a more powerful approach – especially for rare variants.

Response: We agree there are multiple motivations for pooling data. To better reflect this, we revised the first paragraph of the introduction, deleting the unfounded statement about better performance of pooled association analysis. What used to read “Pooled analysis of WGS is useful, due to its computational tractability, ability to control for genetic relatedness across the pooled datasets, and its statistical testing procedures’ improved performance when applied to rare variants.”, now reads “Pooled analysis of WGS is useful due to its computational tractability and its ability to control for genetic relatedness across the pooled datasets.”

2/ Description of the algorithm : the authors should present more clearly the parameters in their algorithm. D, G, M are not previously defined although we understand in passing that this is the number of groups.

2 bis/ I presume that matrix P, using proportion of genotypes within each sub-population is key to the simplification algorithm (factoring on individuals with same genotype within same population). However, for less abstract minds, like the one of your reviewer, I would propose a graph that explains how the genotypes are embedded in the vector/matrix.

Response: in response to these two comments, we extensively re-wrote the description and essentially added another section explaining the logic of the algorithm (pages 9,10, and 11; normally we would copy it here but it is very long). We think it is much clearer now, thank you for asking us to do it!

3/ Simulations : the authors are simulating a difference in Principal Component in order to mimic population genetic heterogeneity. Would it be possible to know which kind of heterogeneity is simulated here – like is it intercontinental for instance ... which Fst could this correspond to ...

Response: There seems to be some confusion here: our use of (simulated) PCs is to provide an indication of how much phenotype variability can be attributed to genotypes, versus other sources. F_{ST} , on the other hand, assesses heterogeneity of genotypes across populations, and so is not directly relevant. Our primary goal with this form of simulation is to mimic the prior work by Musharoff et al.

4/ While the discussion recalls some of the properties of the linear model that might be more relevant in introduction, it fails, for me, in giving a kind of advice on how to proceed. MetaCor looks about right but is too costly, so what are the authors recommendations ? I guess this has something to do with running the most robust of the three methods (excluding MetaCor) and apply the proposed inflation correction ? This should be made clear as this is, according to me, what this paper is about.

Response: We agree and have now expanded the end of the discussion (bottom of page 23) to read: *“Until such methods are developed, we recommend to first use the stratified variance approach, because it is computationally efficient, it can account for relatedness across the entire sample, and the same null model can be used to test variant sets. As a second step, we recommend computing approximate λ_{vS} , and assessing whether observed inflation/deflation remains for test statistics within groups of variants predicted to be inflated/deflated based on λ_{vS} values. If inflation/deflation are observed despite residual variance stratification, the analyst would ideally move forward with a meta-analytic approach such as MetaCor (does not discard data but computationally more demanding), or standard meta-analysis after removing individuals to generate genetically independent strata.”*

REVIEWERS' COMMENTS

Reviewer #2 (Remarks to the Author):

In my opinion the authors have addressed the comments from all reviewers well and this has strengthened the presentation and clarified the message of the paper. I have no further comments.

Reviewer #4 (Remarks to the Author):

I appreciate the efforts of the authors to respond to my queries and comments, and believe they have done so adequately.

Reviewer #5 (Remarks to the Author):

Happy with author's answers